# Latent but not absent: The 'long tail' nature of rural special education and its dynamic correction mechanism

**Bowen Li[1], Guangqin Li[2]\*, Ji Luo[3]\***

**1** Institute for Economic and Social Research, Jinan University, Guangzhou, China, **2** College of International Economics and Trade, Anhui University of Finance and Economics, Bengbu, Anhui, China, **3** School of Public Policy & Management, Tsinghua University, Beijing, China

\* zjfcligq@126.com (GL); luojijiluo2014@hotmail.com (JL)

**Data Availability Statement:** Data has been uploaded to Dryad: https://datadryad.org/stash/share/fKvPUDkLuSyAVvT8XP-JUmg4JVdaActqFcQSoUSNo3E.

## Abstract

The 'long tail' nature of rural special education (RSE) suggests that it simultaneously possesses the private nature of discreteness and the public nature of externalities, which can easily cause provision insufficiency. However, this mismatch may have a dynamic intertemporal correction mechanism impacted by different expenditures of supply sectors (governments and other social sectors). This paper uses different models and data from 30 provinces in China from 2003–2014 to analyze this dynamic correction mechanism. This research finds that different kinds of expenditures from different suppliers have divergent effects on this correction. Capital expenses (especially infrastructure construction) have significantly positive effects on the correction, but administrative expenses have significant dual effects on the correction. These effects may be caused by the various governance efficiencies and motivations of all stakeholders in RSE. This paper concludes that we should pay more attention to the accurate recognition and effective satisfaction of RSE affected by the governance efficiency and motivation of different suppliers to achieve this dynamic correction.

## 1. Introduction

Rural special education (RSE), which is customized to physically or mentally disabled children with heterogeneous demands, constitutes a significant financial commitment and is an important issue for policy and research planning in rural areas [1]. RSE takes place in rural regions, which are vast areas with scattered population distributions, natural disaster risks, and relatively less-advanced economies. Thus, RSE also reflects the characteristics of fragmented distribution, geographical isolation and unbalanced development, along with the relative constraints of financial, community and teaching resources [2–7]. On the other hand, the RSE population is expected to include increasing numbers of students with considerable heterogeneous characteristics [8–10]. Hence, it is easy to cause supply insufficiency, especially in developing countries, and to raise the requirements for its explicit recognition and dynamic correction.

**Funding:** We acknowledge the financial support of the Natural Science Foundation of the Zhejiang Province in China (No. LY18G030014) and the Research Project of Anhui University of Finance and Economics (Green Development Effect of China's Green Credit Policy).

**Competing interests:** The authors have declared that no competing interests exist.

Existing studies often stress that the supply insufficiency of RSE could be explained from multiple perspectives. Many scholars claim that the relatively humble teaching infrastructures and devices in rural areas play dominant roles in causing supply insufficiency of RSE [5,6,11,12]. In addition, RSE faculty members are relatively weak in professional accomplishment and work motivation [2,6,7,13]; they also face lower salaries, mismatched job satisfaction, heavy workloads and vague teaching commitments [14,15]. In addition, affected by the relatively conservative traditional culture and lower income levels, the families and parents of RSE children face barriers to and misuse in covering related expenses and in interacting and mediating with other RSE stakeholders, in addition to lower moral-related motivation and legal awareness with regard to the necessity of RSE for their children [4,16–18]. Furthermore, the public prejudice that stems from a misunderstanding of RSE and the selective absence of community participation increases the severity of the imbalance of RSE [19]. Overall, the combination of the external environmental factors, including the geographical and cultural isolation, along with the intrinsic physical and mental disabilities of RSE children themselves, contributes to RSE being easily ignored and misunderstood.

Within the RSE literature, a longstanding and ongoing criticism of the misuse and low effectiveness of RSE expenditures by different social actors (such as governments, nonprofit organizations and private firms) has always aroused concern in RSE practice. Certain expenditure categories have been particularly controversial for their provision effectiveness. With the exception of Conlin and Jalilevand [20], who consider composition differences and dynamic systematic methods that account for the RSE financing gap, these recent studies have all focused on partial equilibrium analysis to clarify a more standardized way in which the supply insufficiency of RSE is aggregated by factors such as those previously noted. As Conlin and Jalilevand [20] noted, 'much of the empirical work has focused on the financial incentives embedded in local financing environments and how different stakeholders respond to these incentives'. If these disadvantages of RSE children, parents, faculty members and infrastructure explain the challenges to the proper execution of RSE [1], then what is the underlying mechanism and structure of different categories of expenditures that can help to correct the RSE mismatch? Do these financial investments truly have effects? How can we analyze the different roles of expenditures based on the unique characteristics of RSE?.

To answer these questions, several studies have validated underlying concerns addressed by the key roles of specific and special expenditures (or financing and funding), especially their spatial and temporal variations for special education [1,8,21–24]. These studies, however, do not endeavor to determine whether different proportions of RSE expenditures consisting of external and internal elements have diversified marginal effects on the provision of RSE. Few studies have attempted to directly address the primary concern of the correction of RSE imbalance associated with its specified nature in rural areas. No attempt has been made to evaluate whether the heterogeneous expenditure categories might be appropriate for the dynamic recognition and self-correction of RSE insufficiency driven by diversified motivations of different RSE actors.

According to Mahitivanichcha and Parrish [21], there is a complex paradox between the financial incentive and supply implementation of RSE, and this mutual relationship must account for both the horizontal and vertical dimensions (different periods and actors). Several recent studies examine whether the supply insufficiency of RSE would be qualitatively and quantitatively improved by the external development of local economic, social, cultural and cognitive domains [20,25–28]. However, the exact underlying motivation mechanism and the dynamic mode of evolution of RSE insufficiency remain an open question.

Although the concept of RSE has been accepted by most scholars and the general public, the manner in which its intrinsic natures, which are different from those of other education

categories, should be linked to its supply insufficiency and correction is still a major topic of concern. While the embeddedness of the provision and governance structure of RSE has not been given much attention (for a few studies, see Edmonds and Spradlin [28]), in the broader field of general RSE expenditure research, a profound topic of controversy has suggested that to study the mutual interaction and mediation effects of RSE imbalance, we need to consider that the expenditures of other social sectors (as opposed to governments), such as nonprofit organizations (NPOs) and private firms, may exert diversified motivation effects on this self-correction mechanism and subsequently improve the RSE supply conditions in which different social actors operate. Although the supply of RSE has significant gaps in its demand due to the heterogeneity of various suppliers and undertakers [29,30], after precise intertemporal recognition, this trend of supply insufficiency should be ameliorated in light of financial support from different social sectors. Impacted by terrains, cultures and customs in rural areas, RSE should conform to the inner characteristics of rural special children, which is the key to the 'demassification' of public services. The provision insufficiency of RSE may not impact the necessary social securities and needs of the majorities, but the satisfaction of these needs could promote the marginal welfare of local disabled children and conform with the diversified trends of public demands. To understand the intrinsic nature of RSE as the drive to cause supply insufficiency and mismatch, we borrow the concept of the 'long tail' from the field of business management and apply it to comprehending the underlying motivation mechanism of different RSE actors in correcting the mismatch.

As this recent tradition has evolved, the focus has been less on the interaction and mediation of RSE with other education categories, especially general basic education and vocational education, to predict mutual or divergent developmental trends based on the analysis that all of these education categories share institutional environments, economic levels and some demographic variables. In this article, we present key information from the intrinsic nature of RSE and its interaction with other education categories on the topic of RSE correction. We include an overview and theoretical analysis of our conceptual framework and the results of a series of exploratory factor analyses that identified important facets of RSE conditions. The primary focus, however, is on the results and discussion of these interrelationships among different expenditures types, actors and mediation with other education categories.

The conceptual framework used to develop the imbalance and self-correction of RSE was drawn in part from the extant literature on the economics of public service (public good). We extended this research, however, by incorporating the literature on general and dynamic studies as the underlying mechanism to correct RSE supply insufficiency. Existing studies highlight additional information on the problems, development strategies and goals of RSE but seldom further quantitatively analyze when this imbalance occurred and whether, why and how relevant stakeholders have changed their investment approaches and behaviors to correct RSE insufficiency in dynamic dimensions. In fact, due to the pressures of hysteretic recognition, policy quota, public opinion, economic effectiveness and social welfare, different stakeholders of RSE have the impetus to voluntarily adjust their behaviors. Hence, our findings instead have implications for multiple lines of research as follows.

First, we note that despite hundreds of studies on RSE supply and demand, scholars have not noticed the patterns of self-correction and dynamic recognition we document here. Second, our results highlight the importance of expectations about the diversified marginal effects of different expenditures from various stakeholders on RSE. From a more system-equilibrium-oriented perspective, our paper relates to the growing body of research connecting changes and mediation with other basic and vocational education. Finally, identifying and anticipating the adjustment and equilibrium of public goods are themselves some of the most important topics of study in public economics. Our work raises the question of whether RSE, with its

'long tail' nature as both the public good and private good, has an intrinsic mechanism to make its corresponding supply subjects adjust self-behaviors in time. While we leave a thorough evaluation of divergent marginal effects and correction indicators of different RSE expenditures for future work on the premise that all RSE demands are real and reasonable, our results show that these demands deserve consideration, both for forecasting and for understanding how this dynamic disequilibrium achieves self-correction in periods of time fluctuations.

Our work makes several important contributions to the literature. First, our work contributes to both the literature on RSE as well as the literature on (quasi) public goods. We expand the application scope of the 'long tail' in private areas and apply it to RSE first. Prior research has in general considered RSE supply as an outcome of static balance. In this paper, we directly test dynamic correction of RSE as a means of behavior adjustment rather than as a reflection to supply insufficiency by integrating the interactive effect of RSE with other education categories.

Second, our paper is conducive to the RSE literature by examining the beneficial effects of different expenditure categories from various stakeholders. The dominant perspective of the existing literature is that governmental finance is mainly constructive, and therefore, we should highlight its effectiveness more. But our work puts forth a functional view that not only governments but also other social sectors play positive roles in the dynamic correction. Our work finds that RSE has dynamic convergence and correction trends in the next two periods and even more time lags, with improvements in expenditure efficiency, teaching technology, economic levels and social groups.

Third, our work broadens existing literatures of mediation effects among RSE and other education categories. Prior researches has chiefly concerned about how RSE expenditures lead to supply insufficiency itself, whereby RSE expenditures are also determined by other education categories due to the reciprocal relationships among them. On the contrary, our paper accepts a system perspective whereby different stakeholders in all education fields positively engage with affecting the balance of RSE and impacting the diversified marginal effects of RSE. In so doing, we respond to the call for interdisciplinary analysis to consider both supply and demand in the examination of RSE outcomes.

Finally, our work contributes to RSE motivation effects in general. Much of the research on RSE expenditures is premised almost entirely upon empirical studies, which assume that the higher the investment is, the better the effects are, without any subdivided evidence of expenditures. Our work addresses those issues in the literature by splitting expenditure types into infrastructure fees, administrative expenses, welfare allowances to faculty members and scholarships to students. Thus, we advance well-designed paradigms of random experiments and respond to the call for the testing of governance efficiency in RSE with the incorporation of more creative experimental designs in future research.

The rest of the paper is structured as follows. First reviews the relevant background of RSE in China. Second presents the 'long tail' nature of RSE and discusses the theoretical analysis of the underlying mechanism in RSE imbalance and dynamic correction. Third summarizes the data, variables and models used in the estimation. Fourth discusses the regression analysis of the main models and sensitivity analyses. Finally concludes and discusses limitations for future work.

## 2. Background of RSE in China

Special education in China has undergone substantial development and remarkable achievement in quantity, quality and benefit since the reform and opening-up policies in the 1980s.

Based on *the Statistics Bulletin on the Cause of Disabled Persons in China 2017 [China Disabled Persons Federation (2018) 24]*, 43,000 illiterate special children have accepted literacy education in 2017. The enrollment rate of universal education (including nine-year primary- and middle-school education)[1] of visually, intellectually, and hearing-disabled children has surpassed 90% overall. Second, in 2017, 112 high school classes were dedicated to special education nationwide, and 8466 internal students were exclusively enrolled in special education, 7010 of whom were deaf students, and 1456 of whom were blind students. Third, there were 132 secondary vocational schools and 12,968 internal students eligible for special education nationwide in 2017, 3501 of whom were graduate students, and 1802 of whom were students with vocational qualification certificates. In addition, 10,818 special students were enrolled in normal universities and colleges, and 1845 students were enrolled in colleges with higher special education.

However, regarding RSE, the situation is more severe and less optimistic. With 70% of special students living in rural areas, most of these regions face the problem of less-advanced economies and inconvenient transportation. Due to the large disposable sunken cost of construction for RSE schools, most of them are built in counties with relatively concentrated populations. Nevertheless, a considerable proportion of RSE children lack universal education. Based on *Some Opinions on the Development of Special Education (the State Council, 1989)*, China has established the forms of RSE classes attached to normal rural schools, with RSE students 'following-up in class' with other normal students. Currently, the basic pattern of RSE in China is to take a large number of RSE classes as the main body and a certain number of RSE schools as the backbone. Based on the *Outline of the National Medium- and Long-term Education Reform and Development Plan in China [the State Council (2010) 12]*, RSE schools should be completely built in those counties with more than 0.3 million people or large numbers of disabled children. In other counties and rural regions with fewer disabled children in more widespread areas, the overall planning should also be conducted by municipal cities. In addition, intensifying efforts should be made for infrastructure construction and the standardization of the reconstruction of RSE.

From the spatial distribution of RSE in China, the provision and construction gaps between western and central provinces[2] and eastern provinces are still large. Based on some Chinese studies [31], the non-enrollment rate of RSE children in western and central areas could be 27% higher than in eastern areas in China. RSE schools in western and central rural regions face the problems of excessive class size, biased curriculum setting and isolation from residential areas. 'Home-delivery' teaching and long-distance teaching are the better patterns to satisfy the demands of RSE children with scattering distributions and in remote locations. However, only a small proportion of county-run RSE schools implement these teaching services, which made inclusive education difficult to realize in western and central regions.

As to the construction of RSE faculties, the *Opinion on Strengthening the Construction of Special Education Teachers [the Ministry of Education (2012) 12]* first raised the institution of the professional certification of special education faculty members in China. This certification encompasses the combination of teaching and healthcare for RSE children. However, only a few universities and colleges in China have established the major of special education and education rehabilitation, which has led to a major shortage of RSE faculty members.

Besides, the *Measures for the Administration of Special Education Subsidy Funds [the Ministry of Finance and the Ministry of Education (2016) 32]* clearly note the inclusion of special education subsidy funds into general public budgets. This policy could contribute to positive steps in the improvement in RSE financial insufficiency. Based on the *Second Phase of the Special Education Promotion Scheme (2017–2020) [the Ministry of Education (2017) 6]*, the

enrollment rate of universal education for RSE children should be 95% until 2020, thus realizing the comprehensive popularization of RSE.

## 3. The 'long tail' nature of RSE

### 3.1 The definition and characteristics of the 'long tail' nature of RSE

The 'long tail' theory was first raised by Anderson [32] and was primarily used to describe private markets and goods. In this theory, the demand of most private markets and goods was thought to follow a power-law distribution. In the middle of the distribution are popular selling products, that is, the 'head' demand. The satisfaction of the 'head' demand has an economy of scale and scope, which can create huge profits. Both ends of the distribution are 'niche' products with descending demand, that is, the 'long tail' demand. This kind of demand is personalized, scattered, and fragmented, and it lacks an economy of scale and scope. However, this 'long tail' demand will extend along the distribution without disappearance. Along with the development of supply technologies and the entrance of large-scale amateur producers, the marginal cost of supply will decrease to zero, which makes the provision of the 'long tail' products profitable.

We also thought we could use the 'long tail' theory to analyze the relationship among RSE and other education categories, especially general basic education. Analogically, general basic education is similar to the 'head' demand with respect to universality and homogeneity, the satisfaction of which meets economies of scale and scope. RSE is similar to the 'long tail' demand with respect to individualization, fragmentation and heterogeneity. Based on the scattered resources and population distribution in rural areas, we could see the 'long tail' nature of RSE from two perspectives. On the one hand, as a kind of educational category, RSE not only is propitious to the improvement in human capital and the accomplishment of skills among educated individuals [33,34], but also exerts positive influences on local social stability and economic development [35,36]. On the other hand, RSE has some characteristics of private goods due to its utility separability, relative competitiveness and exclusiveness [37,38]. In essence, it still belongs to the domain of public goods. However, compared to other public goods, its publicity has weakened; it could achieve exclusiveness through fee charging; and it has competitiveness among demand subjects, which makes it challenging to scale supply. Hence, the 'long tail' of RSE is the individualized complementation on the general premise of basic education.

The 'long tail' nature of RSE has a progressively decreasing trend of demand amounts and ratios along with increasing demand heterogeneity. The example of China illustrates that from the supply perspective: the numbers of schools, staff, full-time faculty members, entrants and enrolled students for special education in 2016 only require, respectively, 0.4%, 0.3%, 0.3%, 0.3%, 0.1% and 0.1% of all kinds of educational and corresponding indexes. Within special education, the number of rural schools, classes, entrants and enrolled students only hold, respectively, 6.5%, 5.6%, 21.9%, and 22% (not including suburbs and towns) of the whole special education population, which is inconsistent with the vast population distribution of rural areas in China (approximately 50% of the population in rural areas). From the demand perspective, the rate of registered disabled children aged 0–14 years old to the whole population of the same age group was 0.4% in 2016, and the rate in rural areas was 0.3% (calculated by *China Statistical Yearbook*, *China population and Employment Statistics Yearbook*, *China Education Statistics Yearbook and the official website of China Disabled Persons' Federation*). Therefore, RSE requires small percentages from both the supply and demand perspectives. However, within RSE, the student population could be divided into a wide range of categories, including

those with vision, hearing, speech, extremity and intellectual disabilities, which raise differentiated requirements and criteria for their supply.

From the perspective of demand satisfaction cost, due to the territorial imbalance of demand caused by the disproportionate distribution of financial resources, human capital, geographical environments and institutions in rural areas, the supply cost of different RSE (including search costs) is high, which challenges the governance efficiency of supply subjects.

From the perspective of demand levels, the 'long tail' nature of RSE is at a higher level of satisfaction based on Maslow's Hierarchy of Needs [39]. Although the amount of demand is minor from an individual perspective, the whole social welfare (social surplus) of demand satisfaction is large enough to impact the quality of life and happiness for rural citizens. The satisfaction of rural public demands on all levels is the real reflection of the completeness and equality of public services.

From the perspective of the demand driving force, the 'long tail' theory holds that the driving force of private markets is the democratization of production and distribution tools, which binds supply and demand together. The democratization of production tools refers to the participation of grassroots organizational efforts, such as crowdsourcing and crowdfunding, to achieve the decentralization and deauthorization of supply. This theory could also apply to RSE; for example, governments outsource supply services that are difficult for them to satisfy independently to other social subjects, such as NPOs, through service purchases. As rural public services have charitable natures and sociality, even individualized RSE could drive other social participants to satisfy and create social benefits, which is the internal drive of many grassroots groups and charity organizations.

The innovation of the 'long tail' theory is that under specific conditions of organizational operation innovations and information cost control, the scattering and 'demassified' demands (manifested on the curve as the 'long tail') could also arouse more concerns and supply powers of producers. The essence of the 'long tail' demand is to realize the individualized and diversified supplement of the general 'head' demand through group action and real-time coordination and to spread it to the 'long tail' curve end to release distribution bottleneck and information scarcity. Based on the conclusion of the 'long tail' theory, we must further prove whether RSE has the characteristics of the 'long tail' and describe the nature of these characteristics.

### 3.2 The proof of the 'long tail' distribution for RSE

In mathematics, the 'long tail' follows a power-law distribution. The distribution curve of different education categories has a larger slope (faster decrease in speed) and a 'fatter' tail than the standard normal distribution. The curve follows the characteristics of power-law distributions (more accurately, unilateral power-law trailing): the 'head' has sharp changes and few subtypes; the 'tail' has small changes and more subtypes (more with the more; less with the less). In the standard normal distribution, samples are thought to be mutually independent, but in the power-law distribution, samples are thought to be correlated. In fact, different education categories have reciprocal relationships that make the 'long tail' visible.

As Clauset et al. [40] show, we could test the existence of a power-law distribution with the degree of fit of the double logarithmic distribution and the straight line. In the power-law distribution, as shown in Eq (1)

$$y = cx^{-r} \tag{1}$$

where y $>0$, c $>0$, and x $>0$, and take logarithm on both sides of the Eq (1), we will obtain the

Eq (2)

$$Lny = Lnc - rLnx \tag{2}$$

In addition, -r is the slope of the straight line in the double logarithmic distribution. As it is challenging to directly measure the actual demand of all education categories, we could see their distribution from the supply side. We endow all kinds of education categories with rank numbers, from high to low in quantity, and set the logarithmic values of rank order and quantity as the x-axis and y-axis, respectively.

We can check the 'long tail' nature of RSE from the following perspectives. We use the data from the official website of the *Ministry of Education of the People's Republic of China* and the *Statistical Yearbook of China's Educational Expenditure from 2003–2014* (this time period is consistent with those of later empirical studies). First, we checked the school numbers for all education categories. The figures of the distribution curve and the double logarithmic distribution curve of different education categories are shown as follows.

As Fig 1 shows, compared with the standardized normal distribution, the distribution of school numbers is more uneven and appears to have the 'long tail' along the curve. The 'head' of education categories include regular primary schools and preschool education institutions, which exist in large quantities. However, towards the end of the curve, the numbers of special education schools (SES) and other types of schools are small but still extend without disappearance. To further prove the power-law distribution, the double logarithmic distribution of education categories, as shown in Fig 2, closely fits the straight-line trend (due to data constraints, we could not obtain the exact straight line). Hence, we could approximately infer that the distribution of school numbers for education categories follows a power-law distribution and that special education is on the 'long tail' end of the distribution.

A similar trend could also be applied to other indexes, including the number of full-time faculty members and enrolled students in different education categories, as shown below (Figs 3–6).

We could infer that the distribution of different education categories approximately follows the power-law distribution (more data and subdivided indexes allows for greater fit to the power-law distribution). Special education is on the 'long tail' end of the curve. In addition, within special education, we could also find the 'long tail' distribution of RSE due to the

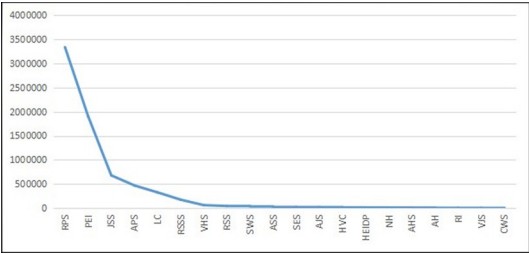

**Fig 1. The distribution curve of school numbers for different education categories in China from 2003–2014 (Unit: Quantity).** Data source are *the Ministry of Education of the People's Republic of China* and the *Statistical Yearbook of China's Educational Expenditure from 2003–2014*. Drawn by STATA 15.0 and Excel 2017. The same below (if no extra description). RPS is Regular Primary Schools. PEI is Preschool Education Institutions. JSS is Junior Secondary Schools. APS is Adult Primary Schools. LC is Literacy Classes. RSSS is Regular Senior Secondary Schools. VHS is Vocational High Schools. RSS is Regular Specialized Secondary Schools. SWS is Skilled Workers Schools. ASS is Adult Specialized Secondary Schools. SES is Special Education Schools. AJS is Adult Junior Secondary Schools. HVC is Higher Vocational Colleges. HEIDP is Higher Education Institutes (HEIs) Offering Degree Programs. NH is Other Nongovernment HEIs. AHS is Adult High Schools. AH is Adult HEIs. RI is Research Institutes. VJS is Vocational Junior Secondary Schools. CWS is Training Schools. The same below.

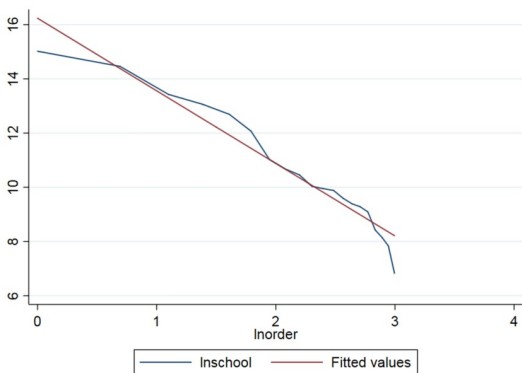

**Fig 2.** The double logarithmic distribution curve of school numbers for different education categories in China from 2003–2014.

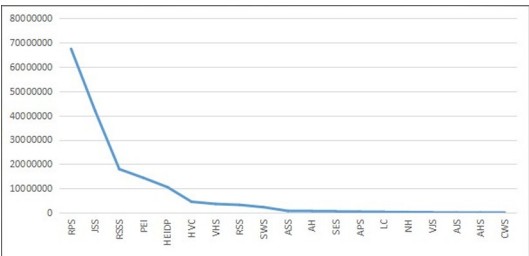

**Fig 3.** The distribution curve of full-time teacher numbers for different education categories in China from 2003–2014 (Unit: Quantity).

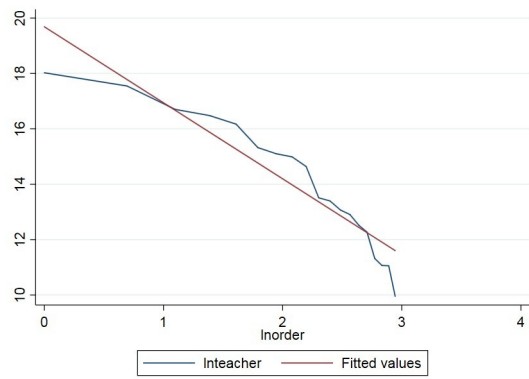

**Fig 4. The double logarithmic distribution curve of full-time teacher numbers for different education categories in China from 2003–2014.**

various categories of special education and uneven geographical allocation. We can calculate the approximate class numbers for RSE based on the proportion of special education in urban, county, town and rural areas, as shown below (Figs 7–9).

In the precise matching of demand and supply, according to the information filtering mechanism of the 'long tail' theory, the inherent defects of the market mechanism inevitably

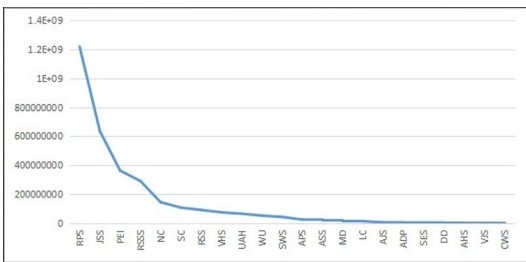

**Fig 5. The Distribution curve of enrolled student numbers for different education categories in China from 2003–2014 (Unit: Quantity).** NC is Undergraduate Courses. SC is Junior College Education. UAH is Undergraduates in Adult HEIs. WU is Network Undergraduates. MD is Master's Degree. ADP is On-the-job Master's Degree. DD is Doctor's Degree. Other indexes are the same as above.

lead to the uneven supply quality of personal products. Anderson [32] thought that recommendation and rating platforms could act as filtering mechanisms to achieve precise supply-demand matching in markets. Likewise, in the demand satisfaction of RSE, governments, as the dominant supply subjects, have their faults in the motivation of the expression of preferences. Their information disadvantages make it challenging to meet the explicit recognition of these various, generally distributed and minor demands thus rendering the supply 'full of noise.' Based on the 'long tail' nature of RSE, its provision can easily cause imbalance and mismatch. Next, we will propose our theoretical analysis and its dynamic correction mechanism on the perspective of different categories of expenditures from different suppliers. We thought that through the dynamic recognition of the specific demand, the multiple participation of RSE provision could act as an information self-filtering mechanism to effectively correct its imbalance.

### 3.3 Theoretical analysis of the dynamic correction for RSE

Set $S_T$ and $D_T$ as the supply and demand of RSE in period T. In the static optimal equilibrium, the following holds tru

$$S_T = D_T \tag{3}$$

$$S_T = EG_T(I_{GT}) + ES_T(I_{ST}) - Ineff_{GT}(I_{GT}) - Ineff_{ST}(I_{ST}) \tag{4}$$

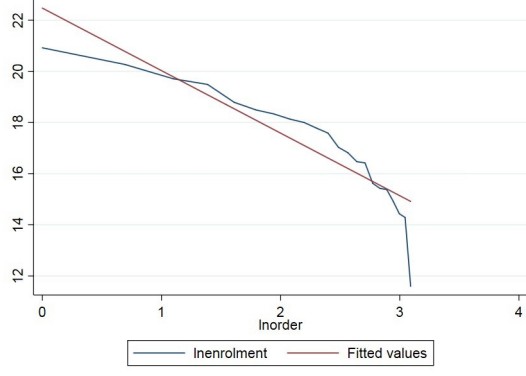

**Fig 6. The double logarithmic distribution curve of enrolled student numbers for different education categories in China from 2003–2014.**

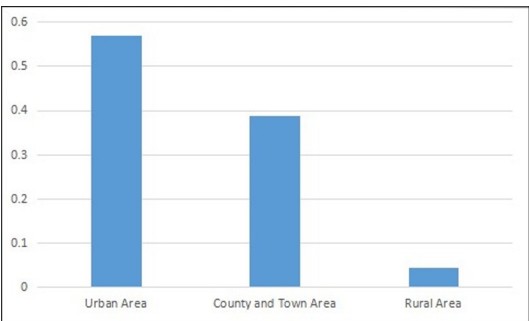

**Fig 7. The geographical proportion of class number for special education (Unit: Proportion).** Data source is *China Education Statistics Yearbook*.

$$D_T = U_{PT}(I_{GT}) + \sum_{i=1}^{N} U_{LT}(I_{ST}) - T_T(I_{GT}) - Fe_T(I_{ST}) \tag{5}$$

In the supply equation, $EG_T$ and $ES_T$ are the utility functions of governments and other social sectors (such as NPOs, private enterprises and individual donations), respectively, for RSE provision in period T. $I_{GT}$ and $I_{ST}$ are the inputs of expenditures from governments and other social sectors, respectively, for RSE in period T. $Ineff_{GT}$ is the function of governance inefficiency for governments in RSE provision. $Ineff_{ST}$ is the function of governance inefficiency for other social sectors in RSE provision.

In the demand equation, $U_{PT}$ is the utility function of the public good nature of RSE for consumers; this function has public visibility and spillover effects. We thought that this variable was more reflected by the input and provision of governments. Based on the maximization of social welfare as a whole, governments are inclined to invest and satisfy those public goods with high homogeneity and scale effects. Hence, $U_{PT}$ is the function of $I_{GT}$. On the other

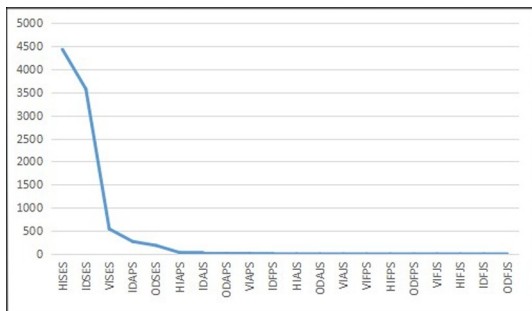

**Fig 8. The distribution curve of class numbers for RSE in China from 2003–2014 (Unit: Quantity).** HISES is Hearing Impairment for Special Education Schools. IDSES is Intellectual Disability for Special Education Schools. VISES is Visual Impairment for Special Education Schools. IDAPS is Intellectual Disability Classes Attached to Primary Schools. ODSES is Other Disability for Special Education Schools. HIAPS is Hearing Impairment Classes Attached to Primary Schools. IDAJS is Intellectual Disability Special Classes Attached to Junior High Schools. ODAPS is Other Disability Classes Attached to Primary Schools. VIAPS is Visual Impairment Classes Attached to Primary Schools. IDFPS is Intellectual Disability Followers in Primary Schools. HIAJS is Hearing Impairment Special Classes Attached to Junior High Schools. ODAJS is Other Disability Special Classes Attached to Junior High Schools. VIAJS is Visual Impairment Special Classes Attached to Junior High Schools. VIFPS is Visual Impairment Followers in Primary Schools. HIFPS is Hearing Impairment Followers in Primary Schools. ODFPS is Other Disability Followers in Primary Schools. VIFJS is Visual Impairment Followers in Junior High Schools. HIFJS is Hearing Impairment Followers in Junior High Schools. IDFJS is Intellectual Disability Followers in Junior High Schools. ODFJS is Other Disability Followers in Junior High Schools.

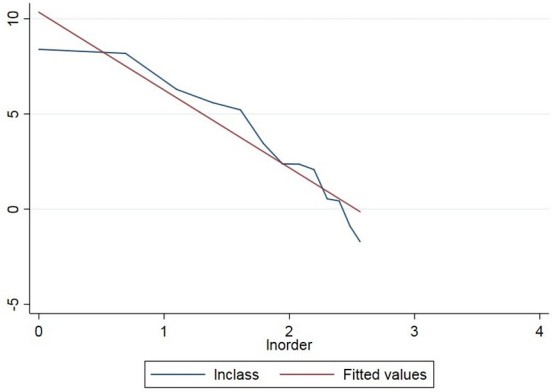

**Fig 9. The double logarithmic distribution curve of class numbers for RSE in China from 2003–2014.**

hand, $U_{LT}$ is the utility function of the 'long tail' nature of RSE for consumers in T, which has superposition property due to its private separability. We thought that $U_{LT}$ was more satisfied and aroused by the input of other social sectors due to the scattering distribution, various categories, and the large number of these organizations as well as their customized services. $N$ is the consumption amount of RSE. $T_T$ is the 'price' (consumers' paid tax) of RSE provided by governments. $Fe_T$ represents the service fees of RSE provided by other social sectors (which could be zero for charitable provisions).

All variables satisfy the following: $\partial EG_T/\partial I_{GT} > 0$,
$\partial ES_T/\partial I_{ST} > 0$, $\partial U_{PT}/\partial(I_{GT}) > 0$, $\partial U_{LT}/\partial(I_{ST}) > 0$, $\partial T_T/\partial(I_{GT}) > 0$, $\partial Fe_T/\partial(I_{ST}) > 0$.
However, this static equilibrium neglects the dynamic fluctuation of RSE. Due to the 'long tail' nature of RSE (such as its preference for concealment and a scattering distribution), it is difficult for suppliers to accurately recognize, locate and effectively provide RSE in time. This issue increases the likelihood for supply insufficiency or the mismatch of supply and demand (mismatch hereafter) for RSE.

When this mismatch has occurred, different suppliers typically have had the impetus and motivation to help correct this imbalance (supply insufficiency) with respect to different factors.

Governments face the requirements of political and governance performance and social reputation to enhance the enrollment rate of RSE or at least limit the unenrollment rate to a minimum standard. Other social sectors (especially NPOs) have the intrinsic nature of altruism and charity to help disadvantaged groups, including uneducated special children. However, it is also difficult for them (especially governments) to invest in RSE provisions and instantly observe effects. This issue indicates a time-lag effect in the impact of the investment in expenditures on RSE correction. In addition, when supplying RSE, the suppliers are based more on the satisfaction of existing needs than only on emerging demands. Therefore, the correction mechanism of RSE provision has time lags compared with that of the emergence of demands. Hence, we could adjust (4) and (5) into a dynamic state:

$$S_{T+1} = EG_T(I_{GT}) + ES_T(I_{ST}) - Ineff_{GT}(I_{GT}) - Ineff_{ST}(I_{ST}) \tag{6}$$

$$D_{T+1} = U_{PT}(I_{GT}) + \sum_{i=1}^{N} U_{LT}(I_{ST}) - T_T(I_{GT}) - Fe_T(I_{ST}) \tag{7}$$

In the dynamic optimal equilibrium, we observe the following:

$$S_{T+1} = D_{T+1} \tag{8}$$

The first-order optimal conditions of $I_{GT}$ and $I_{ST}$ satisfy the following:

$$\partial I_{GT}: \ EG'_T(I_{GT}) + T'_T(I_{GT}) - Ineff'_{GT}(I_{GT}) = U'_{PT}(I_{GT}) \tag{9}$$

$$\partial I_{ST}: \ ES'_T(I_{ST}) + Fe'_T(I_{ST}) - Ineff'_{ST}(I_{ST}) = \sum_{i=1}^{N} U'_{LT}(I_{ST}) \tag{10}$$

Based on (9) and (10), the optimal supply level of RSE for governments in period T+1 depends on the governmental provision capabilities, tax incomes, governance inefficiency and public nature of RSE. On the other hand, the optimal supply level of RSE for other social sectors in period T+1 depends on the social sectors' provision capabilities, consumption fee charging, governance inefficiency and the 'long tail' nature of RSE.

Therefore, the mismatch of RSE has an intertemporal dynamic correction mechanism. When $S_T < D_T$, the suppliers of RSE are inclined to enhance their recognition and provision to make $S_{T+1} \rightarrow D_{T+1}$ and correct the mismatch of RSE.

Considering the different functions and roles of expenditures on RSE provision, we divided the total supply expenditures of governments and other social sectors ($Ex_{GT}$ and $Ex_{ST}$ separately) into four categories: capital expenditure $Cap_T$, administrative expenditure $Adm_T$, welfare expenditure $Wel_T$ and scholarship expenditure $Sch_T$. We believed $Ex_{GT}$ and $Ex_{ST}$ were as follows:

$$Ex_{GT} = Cap_{GT} + Adm_{GT} + Wel_{GT} + Sch_{GT} \tag{11}$$

$$Ex_{ST} = Cap_{ST} + Adm_{ST} + Wel_{ST} + Sch_{ST} \tag{12}$$

$Cap_T$ included the construction and capital expenses for teaching infrastructure, equipment investment (such as computers), and repair for school buildings. These kinds of expenses could enhance the quality of the supply and have relatively stronger positive externalities than other expenses could have, which means that their crowding-out effects would not hamper the supply efficiency and levels. Namely, the following holds true: $I'_{GT}(Cap_{GT}) > I'_{ST}(Cap_{ST}) > 0$.

$Adm_T$ includes hospitality spending, operating expenses and other official service expenses. We thought that a moderate proportion of $Adm_T$ was necessary, but an excessively high proportion may crowd out other supply expenditures and cause resource waste. Hence, we use $Adm_T$ as an index to reflect the provision and governance inefficiency of RSE.

$Wel_T$ includes basic salaries, supplementary wages, welfare payments and bonuses for RSE faculties. It may have double-edged effects on the RSE correction mechanism. From the positive perspective, $Wel_T$ could, on the one hand, be added into the function of provision input as the increase in the quantity of RSE ($I'_{GT}(Wel_{GT}) > 0, I'_{ST}(Wel_{ST}) > 0$). On the other hand, the improvements in welfare and the benefits for RSE faculty members could motivate their work enthusiasm ($Mot_T$ is the function of working motivation). Hence, $Wel_T$ could also affect the quality of RSE provision and be added into the function of motivation ($Mot'_{GT}(Wel_{GT}) > 0, Mot'_{ST}(Wel_{ST}) > 0$). However, from the negative perspective, $Wel_T$ may also crowd out other categories of expenses, such as $Cap_T$, and be adverse to the RSE correction, especially under the distortion and misallocation of $Wel_T$. The route that plays the more dominant role in RSE correction remains to be verified.

$Sch_T$ includes the grants, scholarships and subsidies to RSE students and their families. Compared with other categories of expenditures, the direct beneficiary of $Sch_T$ is RSE students themselves. Hence, $Sch_T$ could affect both the supply and demand equation of RSE at the same

time. From the supply perspective, $Sch_T$ could enhance the degree of satisfaction and help lower the financial barriers of RSE students as one necessary part of provision quality. Hence, $Sch_T$ should be added into the function of provision input ($I'_{GT}(Sch_{GT}) > I'_{ST}(Sch_{ST}) > 0$). From the demand perspective, the improvement in economic treatment may arouse and motivate more unsatisfied potential RSE demand from students' contemporaries and families ($Mos_T$ is the function of demand motivation for RSE children and their families; $Mos'_{GT}(Sch_{GT}) > 0, Mos'_{ST}(Sch_{ST}) > 0$). This mechanism could widen the gap between supply and demand. Hence, the effects of $Sch_T$ that play more dominant roles need to be determined.

In all, in the dynamic correction mechanism of RSE based on different categories of expenditures, the supply and demand equation can be concluded as follows:

$$S_{T+1} = EG_T[I_{GT}(Cap_{GT}, Wel_{GT}, Sch_{GT})] + ES_T[I_{ST}((Cap_{ST}, Wel_{ST}, Sch_{ST})]$$

$$-Ineff_{GT}[I_{GT}(Adm_{GT})] - Ineff_{ST}[I_{ST}(Adm_{ST})] + Mot_{GT}[I_{GT}(Wel_{GT})] + Mot_{ST}[I_{ST}(Wel_{ST})] \quad (13)$$

$$D_{T+1} = U_{PT}[I_{GT}(Cap_{GT}, Wel_{GT}, Sch_{GT})] + \sum_{i=1}^{N} U_{LT}[I_{ST}(Cap_{ST}, Wel_{ST}, Sch_{ST})] - T_T[I_{GT}(Ex_{GT})]$$
$$- Fe_T[I_{ST}(Ex_{ST})] + Mos_{GT}[I_{GT}(Sch_{GT})] + Mos_{ST}[I_{ST}(Sch_{ST})] \quad (14)$$

When $S_{T+1} = D_{T+1}$, the first-order optimal condition could be changed to the following:

$$\partial I_{GT} : U'_{PT}[I_{GT}(Cap_{GT}, Wel_{GT}, Sch_{GT})]$$
$$= EG'_T[I_{GT}(Cap_{GT}, Wel_{GT}, Sch_{GT})] + T'_T[I_{GT}(Ex_{GT})] - Ineff'_{GT}[I_{GT}(Adm_{GT}))$$
$$+ Mot'_{GT}[I_{GT}(Wel_{GT})] - Mos'_{GT}[I_{GT}(Sch_{GT})] \quad (15)$$

$$\partial I_{ST} : \sum_{i=1}^{N} U'_{LT}[I_{ST}(Cap_{ST}, Wel_{ST}, Sch_{ST})]$$
$$= ES'_T[I_{ST}(Cap_{ST}, Wel_{ST}, Sch_{ST})] + Fe'_T[I_{ST}(Ex_{ST})] - Ineff'_{ST}[I_{ST}(Adm_{ST})]$$
$$+ Mot'_{ST}[I_{ST}(Wel_{ST})] - Mos'_{ST}[I_{ST}(Sch_{ST})] \quad (16)$$

Different supply subjects (governments and other social sectors) have various impacts on the correction mechanism of RSE based on their differentiated intrinsic characters and governance efficiency. Social sectors such as NGOs, individual organizations and private enterprises could supply more fragmented and exclusive products based on their advantages of diversification and flexibility, which have lower cost compensation points [41]. However, governments, which serve as the primary undertaker of public service both morally and with respect to duty, are inclined to satisfy more homogeneous demands and place more emphasis on the overall social welfare and economic benefit of provision. Hence, they could apply specific influences on RSE correction by particular expenditure categories. Based on the analysis above, this paper raises the following theoretical hypotheses:

**Hypothesis 1** $Cap_T$ (especially infrastructure construction) has significant positive effects on the dynamic correction of RSE mismatch.

**Hypothesis 2** $Adm_{GT}$ has significant negative impacts on the dynamic correction of RSE. However, $Adm_{ST}$ is not significant due to their minor proportions.

**Hypothesis 3a** $Wel_{GT}$ has significant positive effects on the dynamic correction of RSE. However, $Wel_{ST}$ is not significant.

**Hypothesis 3b** $Wel_{GT}$ has significant negative effects on the dynamic correction of RSE. However, $Wel_{ST}$ is not significant.

**Hypothesis 4a** $Sch_{GT}$ has significant positive effects on the dynamic correction of RSE.

**Hypothesis 4b** $Sch_{GT}$ has significant negative effects on the dynamic correction of RSE.

# 4. Data and methodology

## 4.1 Data

Rural public services in China have undergone significant adjustments since 2003. Chinese central governments have raised the strategy of 'Public Finance Covering the Rural' [42]. Due to data constraints and other factors, this paper uses panel data from 30 provinces in China (excluding Hongkong, Macau, Taiwan and Tibet) from 2003–2014. All data in the models are from the *China Education Statistics Yearbook*, *Statistical Yearbook of China's Educational Expenditure*, *Statistical Yearbook of the Cause of Disabled People in China*, *China Rural Statistics Yearbook*, *China Civil Affairs Statistics Yearbook*, *China Population and Employment Statistics Yearbook*, *China Statistical Yearbook*, the official website of *China Disabled Persons' Federation* and the *Report on the Marketization Index of China's Subprovinces (2003–2016)*. As we discussed in the former theoretical analysis, we divide expenditure categories into capital, welfare, scholarship and administrative expenses. For each category of expenditures, we separate governments from other social sectors (such as NPOs, private firms and individual donations). In addition, considering the dominant role of governments in the construction of teaching infrastructure, we detach infrastructure construction from other capital expenses serving as independent variables. Due to the 'long tail' of RSE, especially its demand concealment, time is required for different expenditures to play roles in recognizing and correcting the mismatch of RSE. Hence, we set the first-order time lag of all kinds of expenditures as the variables to verify the dynamic correction of RSE. Finally, we changed all the missing values to zero.

## 4.2 Methodology

To date, we only consider the dynamic correction mechanism of RSE independently. As a necessary component of the overall education system, RSE has interaction effects with other education categories, especially basic education, which serves as the 'head' of the curve. Notably, the division of the 'head' and 'long tail' in public demand is not absolute and static. With the continuous development of the economy, society and culture, rural citizens have the potential to change the level, quantity and quality of their public demand preferences, which makes the demands of the 'head' and 'long tail' interact and interchange. Hence, we must use system estimation and correct estimation bias. Though the covariant of these education categories (especially the 'head' and 'long tail') have no direct connection, their disturbance term may be related. The seemingly unrelated regression (SUR) can reasonably be used to measure the interaction effects. SUR could also help to correct heteroscedasticity, panel autocorrelation, and contemporaneous correlation (HPAC, see Blackwell [43]).

Considering the 'long tail' nature of RSE, with the dual features of positive externalities as normal education categories and individuality as specific education categories, it is difficult to separate these two attributes to analyze their functions independently. System estimation and SUR give us an implication to address this issue. We thought that to some extent, the externalities of RSE, given its own mismatch (uneducated RSE students), are comparable to the externalities of other 'head' education categories, especially basic education, on the illiteracy rate. From this perspective, we could see the function of different expenditures from basic education (mainly primary and middle school) as the approximate 'proxy variables' on the positive externalities of RSE with its 'long tail' nature. On the relativity, the externality of education

**Table 1. The checklist of all variables.**

| Sign | Name | Calculation |
|---|---|---|
| *Unedu* | The ratio of uneducated RSE children | The registered amount of rural disabled children not enrolled in school / the whole rural population in the age group 0–14 |
| *Illi* | The Illiteracy rate | The whole rural illiterate population/the whole rural population beyond 6 |
| *visual* | The number of uneducated rural visual disabled children | The registered amount of rural visual disabled children not enrolled in school * the rural percentage of population in the age group 0–14 |
| *hearing* | The number of uneducated rural hearing disabled children | The registered amount of rural hearing disabled children not enrolled in school * the rural percentage of population in the age group 0–14 |
| *intellectual* | The number of uneducated rural intellectual disabled children | The registered amount of rural intellectual disabled children not enrolled in school * the rural percentage of population in the age group 0–14 |
| *physical* | The number of uneducated rural physical disabled children | The registered amount of rural physical disabled children not enrolled in school * the rural percentage of population in the age group 0–14 |
| *psychiatric* | The number of uneducated rural psychiatric disabled children | The registered amount of rural psychiatric disabled children not enrolled in school * the rural percentage of population in the age group 0–14 |
| *multi* | The number of uneducated rural multi disabled children | The registered amount of rural multi disabled children not enrolled in school * the rural percentage of population in the age group 0–14 |
| *fiSEcapper* | The capital expenditure of special education from governments (expect infrastructure construction) | The financial expenditure of special education on equipment investment and repair for school building / total fiscal expenditure of special education |
| *soSEcapper* | The capital expenditure of special education from other social sectors (expect infrastructure construction) | The social expenditure of special education on equipment investment and repair for school building / (total expenditure of special education—total fiscal expenditure of special education) |
| *fiBEcapper* | The capital expenditure of rural basic education from governments (expect infrastructure construction) | The financial expenditure of rural basic education (primary and middle school; the same below) on equipment investment and repair for school building / total fiscal expenditure of rural basic education |
| *soBEcapper* | The capital expenditure of rural basic education from other social sectors (expect infrastructure construction) | The social expenditure of rural basic education on equipment investment and repair for school building / (total expenditure of rural basic education—total fiscal expenditure of rural basic education) |
| *fiVEcapper* | The capital expenditure of rural vocational high school from governments (expect infrastructure construction) | The financial expenditure of rural vocational high school on equipment investment and repair for school building / total fiscal expenditure of rural vocational high school |
| *soVEcapper* | The capital expenditure of rural vocational high school from other social sectors (expect infrastructure construction) | The social expenditure of rural vocational high school on equipment investment and repair for school building / (total expenditure of rural vocational high school—total fiscal expenditure of rural vocational high school) |
| *SEconper* | The infrastructure construction expenditure of special education (mainly from governments) | Total expenditure on teaching infrastructure construction / total expenditure of special education |
| *BEconper* | The infrastructure construction expenditure of rural basic education (mainly from governments) | Total expenditure on teaching infrastructure construction / total expenditure of rural basic education |
| *VEconper* | The infrastructure construction expenditure of rural vocational high school (mainly from governments) | Total expenditure on teaching infrastructure construction / total expenditure of rural vocational high school |
| *fiSEadmper* | The administrative expenditure of special education from governments | The financial expenditure of special education on official receptions and operation cost / total fiscal expenditures of special education |
| *soSEadmper* | The administrative expenditure of special education from other social sectors | The social expenditure of special education on official receptions and operation cost /(total expenditure of special education—total fiscal expenditure of special education) |
| *fiBEadmper* | The administrative expenditure of rural basic education from governments | The financial expenditure of rural basic education on official receptions and operation cost / total fiscal expenditures of special education |
| *soBEadmper* | The administrative expenditure of rural basic education from other social sectors | The social expenditure of rural basic education on official receptions and operation cost /(total expenditure of rural basic education—total fiscal expenditure of rural basic education) |
| *fiVEadmper* | The administrative expenditure of rural vocational high school from governments | The financial expenditure of rural vocational high school on official receptions and operation cost / total fiscal expenditures of rural vocational high school |
| *soBEadmper* | The administrative expenditure of rural vocational high school from other social sectors | The social expenditure of rural vocational high school on official receptions and operation cost /(total expenditure of rural vocational high school—total fiscal expenditure of rural vocational high school) |
| *fiSEwelper* | The welfare expenditure of special education from governments | The financial expenditure of special education on basic salaries, supplementary wages, welfare pays and bonuses for teachers / total fiscal expenditure of special education |

*(Continued)*

**Table 1.** (Continued)

| Sign | Name | Calculation |
|------|------|-------------|
| soSEwelper | The welfare expenditure of special education from other social sectors | The social expenditure of special education on basic salaries, supplementary wages, welfare pays and bonuses for teachers / (total expenditure of special education—total fiscal expenditure of special education) |
| fiBEwelper | The welfare expenditure of rural basic education from governments | The financial expenditure of rural basic education on basic salaries, supplementary wages, welfare pays and bonuses for teachers / total fiscal expenditure of of rural basic education |
| soBEwelper | The welfare expenditure of rural basic education from other social sectors | The social expenditure of rural basic education on basic salaries, supplementary wages, welfare pays and bonuses for teachers / (total expenditure of rural basic education—total fiscal expenditure of rural basic education) |
| fiVEwelper | The welfare expenditure of rural vocational high school from governments | The financial expenditure of rural vocational high school on basic salaries, supplementary wages, welfare pays and bonuses for teachers / total fiscal expenditure of rural vocational high school |
| soVEwelper | The welfare expenditure of rural vocational high school from other social sectors | The social expenditure of rural vocational high school on basic salaries, supplementary wages, welfare pays and bonuses for teachers / (total expenditure of rural vocational high school—total fiscal expenditure of rural vocational high school) |
| fiSEschper | The scholarship expenditure of special education from governments | The financial expenditure of special education on scholarship and donors for special students and their families / total fiscal expenditure of special education |
| soSEschper | The scholar expenditure of special education from other social sectors | The social expenditure of special education on scholarship and donors for special students and their families / (total expenditure of special education—total fiscal expenditure of special education) |
| fiBEschper | The scholarship expenditure of rural basic education from governments | The financial expenditure of rural basic education on scholarship and donors for normal students and their families / total fiscal expenditure of rural basic education |
| soBEschper | The scholarship expenditure of rural basic education from other social sectors | The social expenditure of rural basic education on scholarship and donors for normal students and their families / (total expenditure of rural basic education—total fiscal expenditure of rural basic education) |
| fiVEschper | The scholarship expenditure of rural vocational high school from governments | The financial expenditure of rural vocational high school on scholarship and donors for normal students and their families / total fiscal expenditure of rural vocational high school |
| soVEschper | The scholarship expenditure of rural vocational high school from other social sectors | The social expenditure of rural vocational high school on scholarship and donors for normal students and their families/(total expenditure of rural vocational high school—total fiscal expenditure of rural vocational high school) |
| income | The rural per capita income | The rural per capita income/GDP deflator index (ten million RMB per capita) |
| group | The rural per capita amount of social cultural organizations | The amount of rural social cultural organizations / local rural population (10000 per capita) |
| computer | The rural computer room area for special education | Extract directly from raw data ($100km^2$) |
| SEbud | The budget expenditure of special education per RSE student | The total budget expenditure of special education / (total rural student numbers*1000). |
| marketization | Marketization degree | Quote and measure following Fan et al. [44] |
| BEbud | The budget expenditure of basic education per rural normal student | The total budget expenditure of basic education/Total population of rural primary and middle school students |
| rurapop | Rural population | Extract directly from raw data (1 million) |

Due to the limitation of raw data, all kinds of education expenditures in 2012 are absent. We set the mean value of 2013 and 2011 as the substitution value for all kinds of education expenditures in 2012.

(include special and basic education) on them on recipients could be comparable. On the exclusiveness, the effects of expenditure of rural basic education (RBE) have no direct relations on other control variables of RSE. Based on the 'long tail' theory, the underlying mechanism is that heterogeneous 'long tail' demand could aggregate and evolve into mass 'head' demand; the concentrated 'head' demand may also extend along the curve to be individualized 'long tail' demand with the successive satisfaction and transcendence of demands. In this way, we could divide the dual attribute of RSE by different variables. We use different expenditures of RSE as the variables to measure its 'long tail' nature on the disequilibrium and correction. We

**Table 2. The statistic description of all variables.**

| Variable | Obs | Mean | Std. Dev | Min | Max |
|---|---|---|---|---|---|
| unedu | 357 | 7.10702 | 5.215005 | 0.0385852 | 26.56842 |
| illi | 357 | 1.188133 | 2.166472 | 0.0116183 | 18.76768 |
| visual | 357 | 512.9236 | 691.1267 | 0 | 4558.614 |
| hearing | 357 | 572.4153 | 753.1509 | 0 | 4204.157 |
| intellectual | 357 | 998.4957 | 1052.541 | .3038585 | 6148.554 |
| physical | 357 | 927.891 | 982.8844 | .5064309 | 5504.857 |
| psychiatric | 357 | 228.9013 | 275.1201 | 0 | 1601.605 |
| Multi | 357 | 512.7749 | 533.5723 | .1012862 | 2744.542 |
| fiSEadmper | 357 | .1318472 | 0.563777 | 0.102482 | .4101175 |
| soSEadmper | 357 | .2024637 | 1.382536 | 0 | .9098361 |
| fiSEwelper | 357 | .5090991 | .1243886 | .0567159 | .7673993 |
| soSEwelper | 357 | .1665162 | .1592334 | 0 | .7850163 |
| fiSEschper | 357 | .1377826 | .0676744 | .0159894 | .3689919 |
| soSEschper | 357 | .2246669 | .1928869 | 0 | .8429715 |
| fiSEcapper | 357 | .152024 | .1136325 | .0085221 | .8508184 |
| soSEcapper | 357 | .4061389 | .2151026 | 0 | 1 |
| SEconper | 357 | .0568255 | .0984161 | 0 | .6013585 |
| fiBEadmpe | 357 | .1132947 | .0592606 | .006082 | .2917688 |
| soBEadmpe | 357 | .1931151 | .1179527 | .017348 | .5848703 |
| fiBEwelper | 357 | .6531947 | .1520218 | .3820042 | .956934 |
| soBEwelper | 357 | .213845 | .164165 | 0.119222 | .8009072 |
| ifBEschper | 357 | .1063493 | 0.843946 | .0000249 | .3314167 |
| soBEschper | 357 | .2510808 | .2471642 | .000028 | .8578329 |
| ifBEcapper | 357 | .1005923 | .0520529 | 0.170487 | .291999 |
| soBEcapper | 357 | .2869179 | .1420838 | 0.0468524 | .8607732 |
| BEconper | 357 | .0245548 | .0257385 | 0 | .2307642 |
| fiVEadmpex | 248 | .1195169 | .0855835 | 0 | .6554511 |
| soVEadmper | 248 | .2705165 | .1880046 | 0 | 1 |
| fiVEwelper | 248 | .4187416 | .2001796 | 0 | .9911215 |
| soVEwelper | 248 | .184076 | .1779395 | 0 | .7986076 |

use different expenditures of basic education as the approximate 'proxy variables' to measure its social and public nature as educational.

The basic panel data model is as follows in Eq (17):

$$Unedu_{it} = \alpha + \beta_1 FiSEexp_{it} + \beta_2 SoSEexp_{it} + \beta_3 FiBEexp_{it} + \beta_4 SoBEexp + \beta_5 Control_{it} + \mu_i + year_t + \varepsilon_{it} \tag{17}$$

where $i$ represents provinces, and t represents years. *Unedu* is the proportion of uneducated RSE children as dependent variables (DVs). In panel SUR, *Unedu* is also used to measure the illiteracy rate among children in RBE. *FiSEexp* is the proportion of different governmental expenditures for RSE. *SoSEexp* is the proportion of the different expenditures for RSE in other social sectors. *FiBEexp* is the proportion of different governmental expenditures for basic education. *SoBEexp* is the proportion of the different expenditures for basic education in other social sectors. *Control* encompasses other control variables. $\mu$ represents the individual effects of each province (no change over time). *year* represents time effects. $\varepsilon$ is the random error.

The definitions and calculation methods of all variables and their statistical descriptions are listed in Tables 1 and 2.

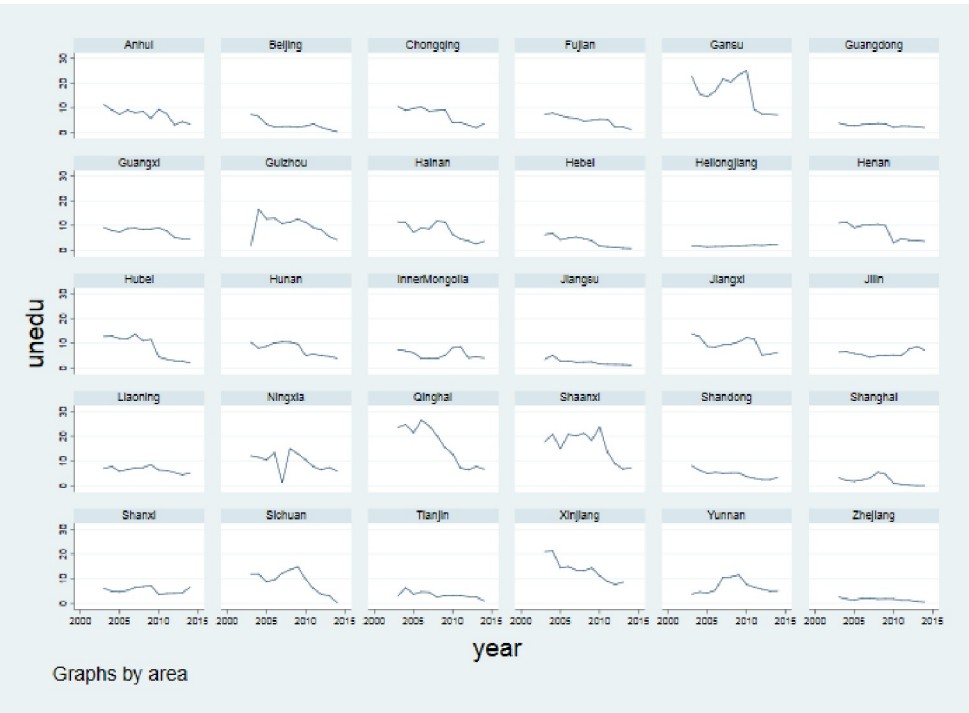

**Fig 10. The time trend of the proportion of uneducated RSE children (*Unedu*) in different provinces of China (2003–2014) (Unit: Proportion).** Data source are *China Rural Statistics Yearbook* and *the official website of China Disabled Persons' Federation.* Drawn by STATA 15.0.

The time trend of *Unedu* in different provinces of China below (Fig 10) shows that different regions vary greatly. Some regions, such as Guangdong, Heilongjiang, Jiangsu and Zhejiang, have very steady trends over time. However, others regions, such as Gansu, Qinghai, Shaanxi and Tibet, have violent fluctuations. Regions could seldom decrease progressively and continually in *Unedu* across this period. Hence, it is necessary to control the variation of each panel as individual effects.

We could also draw the density distribution of each kind of expenditure by different social sectors to check their relative size, as shown below.

The Fig 11 above shows that the expense proportions of infrastructure construction on RSE and rural basic education (RBE) are not high, as we may imagine. Instead, most of these expenses actually account for a small proportion. The largest proportion for mean values is the financial welfare expenditures on RSE. This finding implies that Chinese governments are increasingly concerned about the maintenance and motivation of RSE faculties. In contrast, other social sectors spend less on the welfare of RSE faculty members due to their nature of charity and non-profitability. Further analysis is needed in the next chapter.

## 5. Result analysis

### 5.1 Main model analysis

For the main models of each category of expenditures, we first set normal fixed effects (1) as the base. Then, we consider the interaction of RSE with RBE to set a one-way random effects estimation of panel SUR (2). We put *Illi* as both the independent variables in the first equation and the DV in the second equation (we did not report this in the table). Considering the variance components of the 'long tail' nature of RSE, we use mixed-effects maximum likelihood

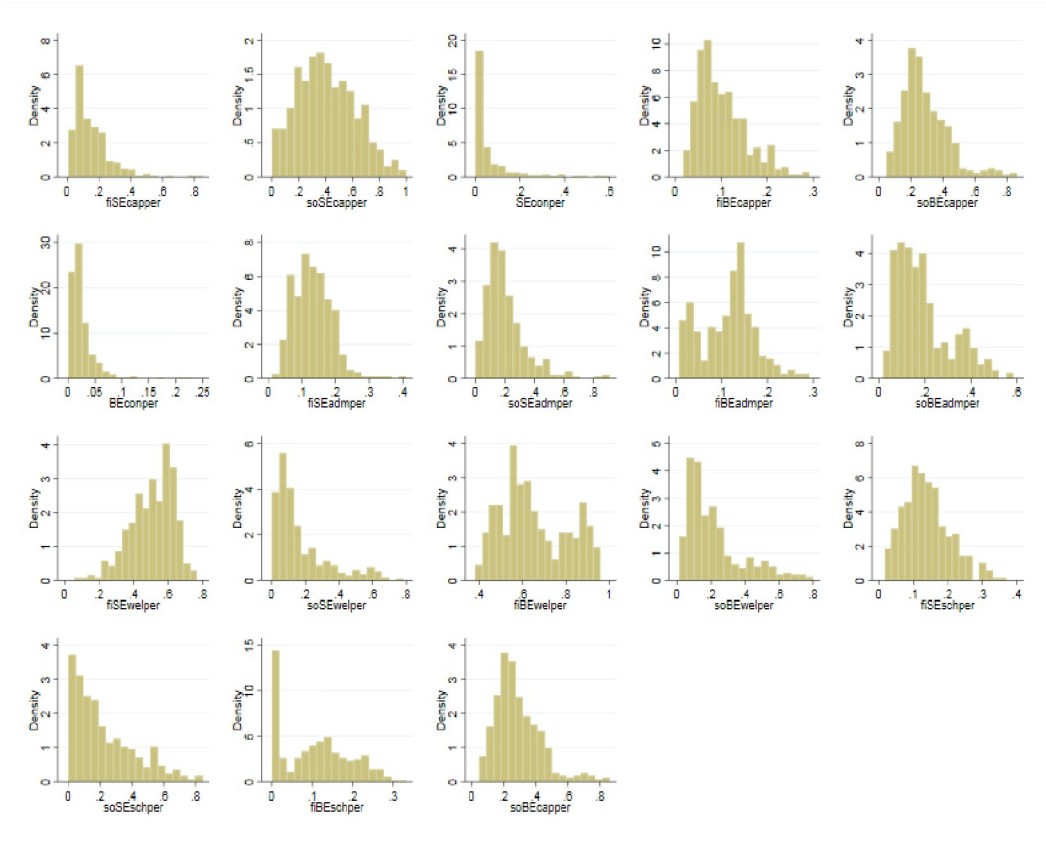

**Fig 11. The density distribution of each kinds of expenditures by different social sectors.** Data source are *China Education Statistics Yearbook, Statistical Yearbook of China's Educational Expenditure*. Drawn by STATA 15.0.

(MEML) (3) to separate the variation of the intercept and coefficients between RSE and RBE [45].

First, the results of capital expense and infrastructure construction are as follows (Table 3):

From the regression results, we found that the coefficients of infrastructure construction (*SEconper* and *Beconper*) are significantly negative, which supports Hypothesis 1 that the infrastructure investment on RSE has large spillover effects on the correction of RSE supply insufficiency. From our hypothesis above, both the public good and the 'long tail' nature of RSE infrastructure investment could help satisfy the heterogeneous RSE demand with high effectiveness in the system equation. In addition, most coefficients of other capital expenses (*fiSEcapper, soSEcapper, fiBEcapper and soBEcapper*) are not stable and significant. In other words, compared with infrastructure construction, other capital expenses, such as the renovation of school buildings and the purchase of other teaching facilities, have smaller marginal effects and may even crowd out other expenditures. The underlying reasons may be due to the failure of signaling for RSE demands, which make trial and error with repetitive investment necessary for its correction.

Second, the results of administrative expenses on the correction are as follows (Table 4):

We found that the coefficients of governmental administrative expenses for RSE (*fiSEadmper*) are significantly positive in most models, which provided support for Hypothesis 2 such that $Adm_{GT}$ has significantly negative impacts on the dynamic correction of RSE. With the 'long tail' nature of RSE, the proportion of administrative expenses reflects the degree of

**Table 3. The results of capital expense on RSE correction.**

|  | (1) | (2) | (3) |
|---|---|---|---|
|  | Fe | Panel SUR | PCSE |
| DV | *Unedu* | *Unedu / Illi* | *Unedu* |
| fiSEcapper | -2.187 | -0.543 | 0.066 |
|  | (-0.888) | (-0.867) | (0.038) |
| soSEcapper | 0.132 | 0.019 | -0.147 |
|  | (0.129) | (0.056) | (-0.162) |
| SEconper | -6.573*** | -3.535*** | -4.111** |
|  | (-3.411) | (-4.774) | (-2.259) |
| fiBEcapper | 1.145 | 16.255*** | 11.037*** |
|  | (0.173) | (9.002) | (2.613) |
| soBEcapper | 2.519 | 5.589*** | 2.518 |
|  | (1.139) | (8.377) | (1.620) |
| BEconper | -43.028** | -13.311*** | -32.183*** |
|  | (-2.146) | (-4.531) | (-3.375) |
| Year | No | Yes | Yes |
| Fixed effect | Yes | No | Yes |
| *N* | 326 | 326 | 326 |

*t* statistics in parentheses.

* $p < 0.1$,

** $p < 0.05$,

*** $p < 0.01$. We didn't show other control variables in the table. The same below.

governance efficiency of RSE: the higher the proportion spent on official receptions and operations costs, the less efficient the governance efficiency. Namely, for a particular education category, it is necessary to maintain the smallest proportion of administrative expenses to serve as the lubricant of RSE implementation. However, what the standard is and how to maintain it remain academically debatable. We can note some clues from the coefficients of governmental administrative expenses for RBE (*fiBEadmper*). Representing the nature of public service in RSE, the financial administrative expense of RBE has positive effects on the correction of

**Table 4. The results of administrative expense on RSE correction.**

|  | (1) | (2) | (3) |
|---|---|---|---|
|  | Fe | Panel SUR | PCSE |
| DV | *Unedu* | *Unedu / Illi* | *Unedu* |
| fiSEadmper | 16.668* | 7.162*** | 8.215 |
|  | (1.909) | (4.934) | (1.562) |
| soSEadmper | -0.767 | 0.096 | -1.098 |
|  | (-0.560) | (0.254) | (-0.744) |
| fiBEadmper | -5.210 | 1.990 | -10.691** |
|  | (-0.859) | (0.920) | (-2.194) |
| soBEadmper | 4.490 | 2.993*** | -0.464 |
|  | (1.369) | (4.723) | (-0.202) |
| Year | No | Yes | Yes |
| Fixed effect | Yes | No | Yes |
| *N* | 326 | 326 | 326 |

**Table 5. The results of welfare expense on RSE correction.**

|  | (1) | (2) | (3) |
|---|---|---|---|
|  | Fe | Panel SUR | PCSE |
| DV | *Unedu* | *Unedu / Illi* | *Unedu* |
| fiSEwelper | 5.836*** | 3.984*** | 2.447 |
|  | (3.279) | (6.162) | (1.023) |
| soSEwelper | 2.255 | 1.255* | -0.263 |
|  | (0.710) | (1.931) | (-0.120) |
| fiBEwelper | 2.410 | -3.947*** | -11.251*** |
|  | (0.694) | (-3.379) | (-3.507) |
| soBEwelper | -0.395 | -2.142*** | -2.545 |
|  | (-0.174) | (-3.127) | (-1.227) |
| Year | No | Yes | Yes |
| Fixed effect | Yes | No | Yes |
| N | 326 | 326 | 326 |

illiteracy. As a kind of pure and basic public good with overwhelming universality and homogeneity, the necessary administrative expense could also act positively on the provision. Hence, the divergent natures of RSE are double-edged and should be discussed separately rather than be examined only from one side. In addition, the coefficients of other social sectors' administrative expenses for RSE and RBE (*soSEadmper* and *soBEadmper*) are not stable, although some of them are significantly negative, especially that of *soSEadmper*. Compared with governments, other social sectors have more flexible organizational and institutional structures, which could meet lower standards of administrative expenses in provision. However, these effects may be overshadowed by the inactive and limited participation of those private organizations due to policy constraints.

Third, the results of welfare expenses on the correction are as follows (Table 5):

We found that the coefficients of governmental welfare expenses for RSE (*fiSEwelper*) are significantly positive, which supports Hypothesis 3b that $Wel_{GT}$ has negative effects on the correction of the RSE mismatch. With the largest proportion of total financial RSE expenditures, their functions of motivation and maintenance for RSE faculty members are blurred and elusive. The underlying reasons may be that the 'long tail' nature of RSE made information on its teaching faculty members and other resources distorted, dissolved and misallocated. Despite the development of RSE infrastructure investment in these years, RSE faculty members still face weak professional accomplishment and planning, as shown in previous studies. However, the opposite is approximately true for the public good characteristic of RSE. The coefficients of governmental welfare expenses for RBE (*fiBEwelper*) at least support the idea that the publicity of RSE still exists and drives welfare effects, even though they are not significantly negative in all models (even the positive coefficients are smaller in absolute value than that of *fiSEwelper*). The coefficients of other social sectors' welfare expenses for RSE and RBE (*soSEwelper* and *soBEwelper*) are still not significant in most models. This finding may be related to the sources of social expenditures. As more social expenditures on RSE are from private donations or corporate profits, they have no legal and moral duties in RSE provision. Based largely in charity and altruism, those welfare expenditures have private attributes and are targeted to specific regions, groups and individuals. Hence, welfare expenses from other social sectors have minor effects on the mismatch.

Finally, we obtain the results of scholarship expenses on the correction, as shown below (Table 6).

**Table 6. The results of scholarship expense on RSE correction.**

|  | (1) | (2) | (3) |
|---|---|---|---|
|  | Fe | Panel SUR | PCSE |
| DV | *Unedu* | *Unedu / Illi* | *Unedu* |
| fiSEschper | 5.063 | -0.234 | 3.381 |
|  | (0.641) | (-0.173) | (0.768) |
| soSEschper | -1.376 | -1.039** | -2.074 |
|  | (-0.671) | (-2.246) | (-1.249) |
| fiBEschper | 0.769 | 7.371*** | 1.356 |
|  | (0.137) | (3.863) | (0.321) |
| soBEschper | -2.290 | -1.358** | 0.196 |
|  | (-1.616) | (-2.231) | (0.147) |
| Year | No | Yes | Yes |
| Fixed effect | Yes | No | Yes |
| *N* | 326 | 326 | 326 |

The results show that the coefficients of both governmental scholarship expenses for RSE and RBE (*fiSEschper* and *fiBEschper*) are not significant in most models, which did not support either Hypothesis 4a or Hypothesis 4b and obscured the double-edged sword. However, the absolute value of *fiBEschper* showed overwhelmingly smaller coefficients than those of *fiSEschper*, thus reminding us to see this issue from the dual 'long tail' nature of RSE. At least the nature of the public good in the form of basic education is rooted in sufficient rationality to cover more financial burdens for uneducated RSE children. However, the 'long tail' scattering of RSE distorts distribution and creates information asymmetry. Especially in poor rural areas, the high opportunity cost and unclear value of education led to many families being unaware of the need to receive and support education and unwilling to receive aid for their special children, even when facing rich donors.

This dampening mechanism could be given more support from the coefficients of other social sectors' scholarship expenses for RSE and RBE (*soSEschper* and *soBEschper*). The 'massive amateur producer [32]' represented by numerous and various NPOs could directly interact with citizens' real demands in the form of grassroots and the 'long tail aggregator'. They could collect and filter information through platform sharing and have exclusive and accurate donors for RSE students. Hence, the coefficients of *soSEschper* and *soBEschper* are significantly negative in most models. These results are consistent with those of Anderson [32], indicating that the aggregation of the 'long tail' acts as the democratic distribution. This aggregation effect is the most significant in scholarship expenses because this category of demand could exert the information and cost advantages best, as it essentially faces the exact RSE demanders.

## 5.2 Sensitivity and mechanism analysis

The sensitivity and mechanism analysis will be divided into four parts. First, due to the variety and complexity of different disability types, public expenditures may have differentiated influences on their RSE mismatch and correction. Hence, we separate RSE children into those with visual disability, hearing disability, speech disability, physical disability, intellectual disability, psychiatric disability and multiple disabilities. Furthermore, the interaction effects of RSE with other categories are not only between special and basic education. Professional education, as an alternative path, also has mediation influences as the 'head' on the 'long tail' of RSE. Thus, we set the system estimation of RSE and professional education instead. Third, due to the change in the policy implementation of RSE in 2009, we view the policy as a quasi-random

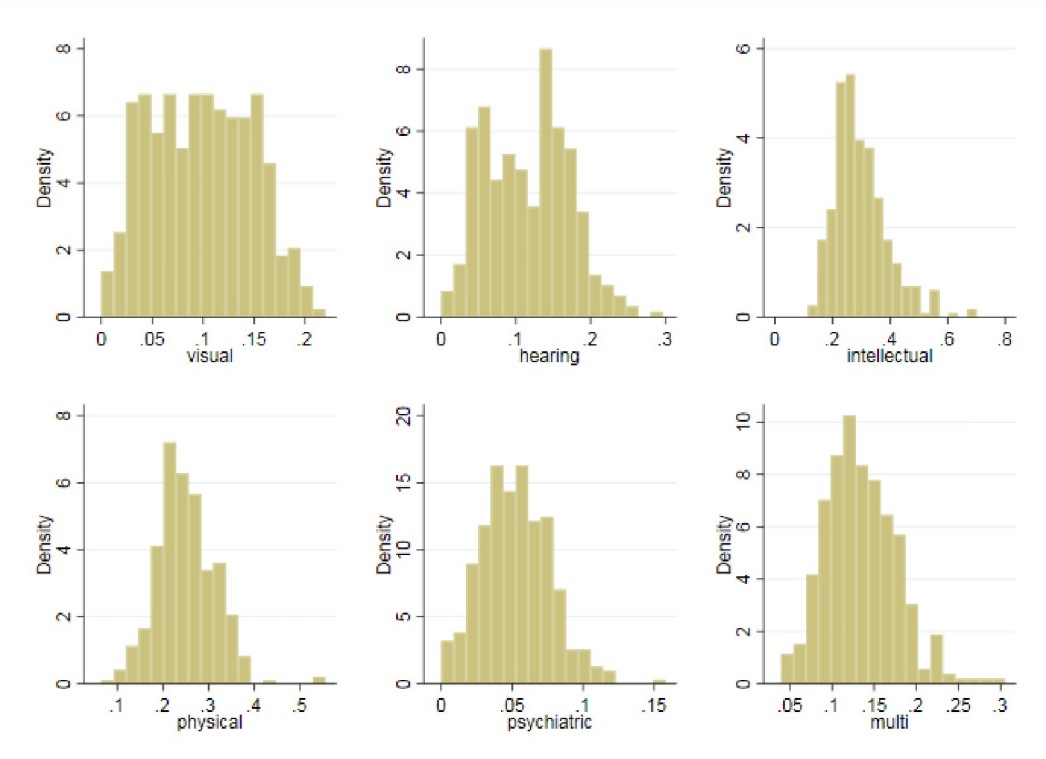

**Fig 12. The proportion distribution of different sub-types of RSE.** Data source is *the official website of China Disabled Persons' Federation*. Drawn by STATA 15.0.

experiment to perform difference-in-differences (DID) and a synthetic control method (SCM) of public expenditures on RSE as a robustness check. Finally, we test the time dynamic effect of the correction mechanism by changing independent variables into current periods and second-order time lags.

**5.2.1 The sensitivity check of different expenditures on the subtypes of RSE correction.**   Due to the 'long tail' nature of RSE, its demand is fragmented and individualized with various symptoms and disabilities. Hence, it is necessary to analyze its correction mechanism in specific subtypes. Based on the official website of the *China Disabled Persons' Federation*, we divide RSE children into categories characterized by 6 kinds of disabilities: visual, hearing, physical, intellectual, psychiatric and multi disabilities. From the distribution of the proportions of different subtypes of RSE, as shown below (Fig 12), we found that the proportions of RSE do have differentiated degrees of prevalence. Intellectual disabilities account for the largest proportion of the whole population of students in RSE, which proves that these disabilities are widespread. However, visual and psychiatric disabilities are relatively minor in all subtypes.

We use the MEML method to measure the correction of different expenditures on these subtypes of RSE. We choose the total number of uneducated RSE children for each subtype as the DV for a robustness check. The results of capital expenditures on subtypes of RSE correction are shown below (Table 7).

We found that for infrastructure investment and construction, the 'long tail' nature of different subtypes of RSE on the correction is still robust (the negative significance of *SEconper*). In addition, the 'head' nature of RBE (*BEconper*) is also negative on the correction, although some of them are not significant. Regarding different subtypes of RSE, we found that the

**Table 7. The results of capital expenditures on subtypes of RSE correction.**

|  | (1) | (2) | (3) | (4) | (5) | (6) |
|---|---|---|---|---|---|---|
|  | visual | hearing | physical | intellectual | psychiatric | multi |
| fiSEcapper | -360.859* | -407.691** | -198.023 | -212.250 | -100.116 | -89.171 |
|  | (-1.918) | (-2.105) | (-0.855) | (-0.786) | (-1.202) | (-0.619) |
| soSEcapper | -78.173 | -51.721 | 51.952 | 197.948 | -26.870 | -55.894 |
|  | (-0.728) | (-0.470) | (0.390) | (1.288) | (-0.578) | (-0.679) |
| SEconper | -820.271*** | -903.147*** | -596.098*** | -873.989*** | -312.116*** | -418.977*** |
|  | (-4.371) | (-4.668) | (-2.585) | (-3.242) | (-3.735) | (-2.919) |
| fiBEcapper | 353.128 | -511.589 | 603.822 | 175.714 | 130.264 | 264.414 |
|  | (0.708) | (-1.004) | (0.972) | (0.247) | (0.608) | (0.691) |
| soBEcapper | 406.943* | 786.528*** | 501.455* | 767.286** | 105.889 | 131.938 |
|  | (1.940) | (3.683) | (1.913) | (2.563) | (1.195) | (0.819) |
| BEconper | -2.1e+03** | -1.4e+03 | -2.3e+03* | -550.462 | -821.156** | -1.2e+03 |
|  | (-2.150) | (-1.420) | (-1.919) | (-0.396) | (-1.962) | (-1.560) |
| N | 326 | 326 | 326 | 326 | 326 | 326 |

We didn't show other control variables in the table. The same below.

absolute values of the coefficients of psychiatric and multiple disabilities are smaller than those of other subtypes. Compared to other disabilities, psychiatric children have intrinsic difficulties in education. Some of these individuals face much higher education costs and are enrolled in full-time daycare. Hence, for these children, there is a lack of service availability and motivation even with the promotion of capital investment. On the other hand, we can see that multiple disabilities represents a comprehensive category that has mediation effects on other subtypes of disabilities. Accordingly, the coefficients for children with multiple disabilities are in the range of the coefficients of other disabled children with regard to absolute value. In addition, the coefficients of *visual* and *hearing* are most significantly negative in all models (*fiSEcapper* and *soBEcapper*). Most visually and hearing-disabled children have the capacity to study and be educated if they could receive suitable auxiliary teaching equipment to help them overcome physical inconveniences. In China, most of these children are being educated independently in special schools for individuals who are deaf and mute. However, in rural areas, the necessary auxiliary teaching equipment could hardly be fully satisfied, which is the underlying reason why capital expenses have larger marginal effects on visually and hearing-disabled RSE children.

We discuss the results of administrative expenditures on subtypes of RSE correction below (Table 8).

We found that consistent with the main models, the 'long tail' nature of different RSE still causes administrative expenses play separate roles in RSE and RBE (the positive significance of *fiSEadmper* and the negative significance of *fiBEadmper*). A moderate range of administrative expenses is necessary, especially considering the attribute of RSE as a public service such as RBE. However, the excessive burden of administrative expenses makes the scattering distribution of the 'long tail' difficult to satisfy and hampers its recognition. Regarding the different subtypes of RSE, the coefficients of *psychiatric* and *multisubtypes* are still smaller in absolute value. Besides, the insignificance of the coefficients of physical disabilities is noteworthy. RSE children with physical disabilities could still compete equally with their normal RBE counterparts except for movement inconvenience. In China, a considerable proportion of physical disabled children (such as crippled children) are being educated in the pattern of 'follow-up in

**Table 8. The results of administrative expenditures on subtypes of RSE correction.**

|  | (1) | (2) | (3) | (4) | (5) | (6) |
|---|---|---|---|---|---|---|
|  | visual | hearing | physical | intellectual | psychiatric | multi |
| fiSEadmper | 1045.185** | 981.101** | 719.817 | 144.129 | 439.513** | 448.898 |
|  | (2.242) | (2.150) | (1.292) | (0.234) | (2.205) | (1.298) |
| soSEadmper | 79.822 | 35.320 | -76.520 | -204.264 | 48.315 | -1.012 |
|  | (0.607) | (0.275) | (-0.493) | (-1.189) | (0.847) | (-0.010) |
| fiBEadmper | -1.4e+03** | -3.1e+03*** | -2.4e+03*** | -3.9e+03*** | -763.666*** | -1.3e+03*** |
|  | (-2.576) | (-5.735) | (-3.675) | (-5.437) | (-3.331) | (-3.256) |
| soBEadmper | 315.218 | 717.855*** | 64.879 | 744.778** | -10.462 | -94.645 |
|  | (1.375) | (3.187) | (0.234) | (2.439) | (-0.109) | (-0.554) |
| N | 326 | 326 | 326 | 326 | 326 | 326 |

class' with other normal students. Under this situation, they may be the least affected RSE group among all subtypes.

We discuss the results of welfare expenditures on subtypes of RSE correction below (Table 9).

In contrast to the main model, a positive significance of *fiBEwelper* was found in some models. This finding implies that with the subdivision of RSE categories, the public good nature of the positive motivation effects for RSE faculty members will fade and shift to its 'long tail'. Especially for psychiatric and multidisabled children, the mentality and chronicity of their disabilities, along with the mediation effects, make it easy for RSE faculty members to deem their efforts to be fruitless and discouraging. Finally, we turn to the results of scholarship expenditures on subtypes of RSE correction, as shown below (Table 10).

We found that most coefficients of scholarship expense are still not significant. However, the coefficients of *soBEschper* become significantly negative. This finding supports our ideas that other social sectors such as NPOs can customize and lock onto specific recipients with the largest marginal benefit of scholarships due to their information advantage. Although the whole proportion of *soBEschper* is not significant in the main models, with the extension of the 'long tail', heterogeneous demands will be more apt to other social actors as the 'long tail aggregator'.

In summary, different subtypes of RSE exert divergent effects on the correction mechanism. Regarding visual and hearing disabilities, the best solution is to highlight the investment in

**Table 9. The results of welfare expenditures on subtypes of RSE correction.**

|  | (1) | (2) | (3) | (4) | (5) | (6) |
|---|---|---|---|---|---|---|
|  | visual | hearing | physical | intellectual | psychiatric | multi |
| fiSEwelper | 927.478*** | 874.023*** | 631.579*** | 650.380*** | 305.490*** | 437.080*** |
|  | (5.528) | (5.187) | (3.099) | (2.802) | (4.105) | (3.444) |
| soSEwelper | 47.475 | -22.523 | -68.436 | -169.921 | -24.646 | -99.522 |
|  | (0.250) | (-0.118) | (-0.296) | (-0.646) | (-0.295) | (-0.692) |
| fiBEwelper | 185.212 | 1037.818*** | 640.588** | 1490.802*** | 63.150 | 94.182 |
|  | (0.841) | (4.689) | (2.377) | (4.873) | (0.660) | (0.565) |
| soBEwelper | 63.663 | 48.273 | 112.543 | -11.502 | 72.742 | 272.585** |
|  | (0.353) | (0.267) | (0.513) | (-0.046) | (0.918) | (1.998) |
| N | 326 | 326 | 326 | 326 | 326 | 326 |

**Table 10. The results of scholarship expenditures on subtypes of RSE correction.**

|            | (1)         | (2)          | (3)          | (4)          | (5)         | (6)         |
|------------|-------------|--------------|--------------|--------------|-------------|-------------|
|            | visual      | hearing      | physical     | intellectual | psychiatric | multi       |
| fiSEschper | 273.721     | 189.506      | 61.280       | 429.870      | 152.412     | -186.051    |
|            | (0.558)     | (0.385)      | (0.100)      | (0.634)      | (0.776)     | (-0.502)    |
| soSEschper | -9.251      | -22.372      | 4.232        | -230.510     | 52.291      | 61.631      |
|            | (-0.055)    | (-0.133)     | (0.021)      | (-1.007)     | (0.768)     | (0.491)     |
| fiBEschper | -313.008    | -982.078**   | -1.0e+03**   | -2.2e+03***  | -74.456     | 156.515     |
|            | (-0.733)    | (-2.292)     | (-1.976)     | (-3.895)     | (-0.408)    | (0.495)     |
| soBEschper | -283.257**  | -559.192***  | -325.121**   | -517.422***  | -88.747*    | -216.916**  |
|            | (-2.419)    | (-4.756)     | (-2.346)     | (-3.322)     | (-1.751)    | (-2.507)    |
| N          | 326         | 326          | 326          | 326          | 326         | 326         |

auxiliary teaching facility member to help promote the availability of RSE to children with such disabilities. Regarding physical disabilities, the existing pattern of 'follow-up in class' could help them enjoy equal opportunities to access RBE without any artificial entrance barriers. With respect to individuals with intellectual and psychiatric disabilities, due to the relatively higher entrance costs of education, they are less affected by expenses from governments and other social sectors. Multiple disabilities could be seen as a comprehensive category and as a method of mediation for other subtypes of disabilities; thus, their marginal effects are also moderate. In addition, due to the 'long tail' nature of specific subtypes of RSE, other social sectors could provide more assistance in locating, matching and sponsoring the accuracy of unsatisfied RSE demands as the correction mechanism in those subfields.

**5.2.2 The interaction and mediation effects of RSE with Rural Vocational Education (RVE0).** To date, we only consider the interaction and mediation effects of RSE and RBE to act similarly to the complementation between the 'head' and the 'long tail' in the power-law distribution. Due to the intrinsic physical or mental disadvantages of RSE children, it is sometimes more important for them to master vocational skills for their livelihood rather than to acquire comprehensive knowledge. Hence, it is necessary to consider the role of rural vocational education (RVE), which acts as a substitution for or a competing option against RSE, especially the public good nature of RSE. However, RBE is different because RVE is also at the 'long tail' end of the power-law distribution of all education categories (see higher vocational colleges (HVCs) in Fig 1). Therefore, the interaction and mediation effects of RSE and RVE will be less complementary but more competitive. To prove this statement, we set the different proportions of different expenditures on RVE (HVCs) as the alternative 'proxy variable' instead of testing RBE. We still set the fixed-effects, FGLS, PCSE and MEML models. In addition, we changed all of the missing values to zero. Due to the hard convergence of panel SUR between RSE and RVE variables, we ignore the panel SUR model. The results of capital expenses on the correction are presented below (Table 11).

We found that the coefficients of *SEconper* were still significantly negative, which supports our idea that the expense of infrastructure construction has the largest marginal effect on the correction of RSE. However, this time, the coefficients of RVE (*fiVEcapper*, *soVEcapper*, *VEconper*) are no longer significant. This finding is consistent with the interaction between RSE and RVE: both are on the 'long tail' end of the curve. Hence, it proves they are more competitive than complementary relations, which means that both show a homogeneous 'long tail' nature. Then, we turn to the administrative expenditures on RSE correction with RVE, as shown below (Table 12).

**Table 11. The results of capital expenditures on RSE correction with RVE.**

|  | (1) | (2) |
|---|---|---|
|  | Fe | PCSE |
| DV | *Unedu* | *Unedu* |
| fiSEcapper | -3.075 | 0.142 |
|  | (-1.110) | (0.076) |
| soSEcapper | 0.200 | 0.437 |
|  | (0.162) | (0.484) |
| SEconper | -8.077*** | -4.572** |
|  | (-3.563) | (-2.466) |
| fiVEcapper | -0.984 | 0.570 |
|  | (-0.337) | (0.340) |
| soVEcapper | 1.067 | 0.795 |
|  | (1.046) | (0.857) |
| VEconper | -1.962 | 5.540** |
|  | (-0.508) | (2.128) |
| Year | No | Yes |
| Fixed effect | Yes | Yes |
| *N* | 236 | 236 |

Due to data limitation, we could only get the data of RVE from 2007–2014. The same below.

We found that only the coefficients of *fiVEadmper* are significantly negative in these models. This finding indicates that the moderate investment of administrative expenses will not harm the positive spillover effects of RSE; instead, it helps to correct the mismatch, as other education categories participate. In addition, the coefficients of *fiSEadmper* are still significantly positive, which is consistent with the main models above. Next, we analyze the results of welfare expenses on RSE correction with RVE (Table 13).

We found that the coefficients of *fiSEwelper* were still significantly positive. This finding implies that even when the substitution of RVE for RBE was considered, the motivation effects

**Table 12. The results of administrative expenditures on RSE correction with RVE.**

|  | (1) | (2) |
|---|---|---|
|  | Fe | PCSE |
| DV | *Unedu* | *Unedu* |
| fiSEadmper | 16.371* | 8.713* |
|  | (1.963) | (1.671) |
| soSEadmper | -1.630 | -2.617* |
|  | (-1.062) | (-1.933) |
| fiVEadmper | -6.654 | -6.576** |
|  | (-1.148) | (-2.245) |
| soVEadmper | 1.110 | 0.151 |
|  | (0.827) | (0.105) |
| Year | No | Yes |
| Fixed effect | Yes | Yes |
| _cons | 11.504*** | 15.933*** |
|  | (3.109) | (8.463) |
| *N* | 236 | 236 |

**Table 13. The results of welfare expense on RSE correction with RVE.**

|  | (1) | (2) |
|---|---|---|
|  | Fe | PCSE |
| DV | *Unedu* | *Unedu* |
| fiSEwelper | 6.159*** | -1.188 |
|  | (3.507) | (-0.494) |
| soSEwelper | 2.570 | -3.154 |
|  | (1.012) | (-1.468) |
| fiVEwelper | 2.565 | -1.156 |
|  | (1.100) | (-0.663) |
| soVEwelper | -3.231* | -1.112 |
|  | (-1.773) | (-0.664) |
| Year | No | Yes |
| Fixed effect | Yes | Yes |
| _cons | 8.386* | 17.409*** |
|  | (2.013) | (7.159) |
| N | 236 | 236 |

of welfare expenses on RSE faculty members were still distorted and misleading. As Koutrouba et al. [12] argued, the considerable hesitation and prejudice from RSE families and societies are still widespread and even affect the development and inclusion of RVE. The natural feelings of inferiority and judgment of vocational education in China made RSE and RVE faculty members prefer to stay in comprehensive colleges [46,47]. We can also see a similar situation for the results of scholarship expenses on RSE correction with RVE, as shown below (Table 14). This time, the coefficients of neither *fiSEschper* nor *fiVEschper* were significant. The weak efficacy of welfare and scholarship on either faculty members or students proves that the mismatch of RSE is not the direct result of insufficient investment. There are more complex underlying reasons related to keeping all stakeholders financially and spiritually motivated.

In all, the interaction of RSE with RVE is not as functional as that with RBE due to the sharing 'long tail' nature of RSE and RVE. However, the main regression results still support our

**Table 14. The results of scholarship expense on RSE correction with RVE.**

|  | (1) | (2) |
|---|---|---|
|  | Fe | PCSE |
| DV | *Unedu* | *Unedu* |
| fiSEschper | 8.238 | 2.224 |
|  | (1.227) | (0.434) |
| soSEschper | -0.757 | -1.465 |
|  | (-0.303) | (-0.784) |
| fiVEschper | 0.221 | -0.702 |
|  | (0.114) | (-0.423) |
| soVEschper | -2.369 | -1.630 |
|  | (-0.731) | (-0.757) |
| Year | No | Yes |
| Fixed effect | Yes | Yes |
| N | 236 | 236 |

hypothesis that capital expenses (mainly infrastructure construction) have the largest marginal effects on the correction. In addition, administrative expenses have dual effects on the dual nature of RSE. For its 'long tail' nature, they could dampen the governance efficiency of RSE provision. For its nature of public goods, a moderate proportion of administrative expenses is necessary for the normal operation of the correction mechanism. The other two categories (welfare and scholarship) of expenses have obscure and blurry effects on the correction due to distortion and misallocation.

## 5.3 Implication

Generally, due to the time-lag recognition and provision of RSE, different categories of expenditures from various suppliers have divergent influences on the intertemporal correction mechanism. In detail, capital expenses (especially infrastructure construction) have the largest positive marginal effects on the dynamic correction of RSE mismatch. This finding is true for both the 'long tail' and the public nature of RSE and considering its interaction with other education categories.

Second, governmental administrative expenses have significant negative impacts on the dynamic correction of RSE. However, the administrative expenses of other social sectors are not significant due to their minor proportions. We can see the proportion of administrative expenses as the indicator of governance efficiency in the provision. For the public nature of RSE, similar to other education categories, a moderate proportion of RSE is necessary for its normal operation. However, with regard to its 'long tail' nature, the excessive investment of administrative expenses could crowd out other kinds of expenses and have side effects on the correction. Hence, related policies should consider such an investment more seriously and control it reasonably.

Third, the influences of welfare expenses for faculty members from both governments and other social sectors were not significant for the dynamic correction of RSE. The underlying reason may be the distortion and misallocation of these resources into the weak motivation of RSE faculty members. Due to the institutional, cultural and social barriers they face, RSE faculty members are still insensitive to financial incentives due to their blurry career expectations.

Fourth, the influences of scholarship expenses for students and their families from both governments and other social sectors were also not significant for the dynamic correction of RSE. In addition to the similarity in the motivation of faculty members with regard to welfare expenses, the cognitive deficiency of RSE families and the high entry cost of education also decrease the effects of financial investments directly to the families. However, the aggregation of the 'long tail' could act as a democratic distribution and exert the information and cost advantage best, as it essentially faces the exact RSE demanders.

In the sensitivity analysis, we found that children with psychiatric disabilities were less sensitive to financial investments than children with other physical disabilities (including vision- and hearing-related). The underlying mechanism is that children with psychiatric disabilities have congenital disadvantages to being educated and are more inclined to receiving healthcare than education. In addition, the marginal effects on multi-disabled children are the mediation of children with physical and psychiatric disabilities.

In the test of mediation of RSE with RVE, the results were not as functional as those with RBE due to the sharing 'long tail' nature of RSE and RVE. The 'long tail' nature could dampen the governance efficiency of RSE provision. This finding implies that most RSE children in China are accepting basic education rather than vocational education. Hence, the relationship between RSE and RVE is not complementary but rather continuous on the 'long tail' curve.

Finally, the policy evaluation of infrastructure construction in the western and central regions of China in 2009 support the significance of capital expenses as the chief task to correct

the supply insufficiency of RSE. The verification of the time dynamic effect supports the idea that this recognition and correction have time-lag properties. Hence, the investment of RSE expenditures must consider this utility lag and place greater emphasis on the improvement of existing RSE conditions.

## 6. Conclusion and limitations

The explanatory model and theories developed in this paper are convincing since they were replicated in different subtypes of RSE with different measuring methods and mediation effects. These findings should be viewed as part of an evolving understanding of RSE conditions for its dynamic correction mechanism based on Conlin and Jalilevan [20] and the more general discussion of financing disabled children [23], representing close to a decade's worth of research.

Two of the constructs that emerged from our paper seem particularly important. First, conceptualizing the 'long tail' nature of RSE and its interaction and mediation with other 'head' services, such as RBE, appears to be more useful. A second important finding is that the dynamic correction mechanism of RSE will be affected by different expenditures due to the differentiated function and motivation of various social actors. This mechanism constitutes a highly salient construct that emerged from the analysis of Ho and Chen [48] in which the costs and fiscal incentives for special education vary significantly according to the severity and variety of disability. This paper has much more clearly defined the broader constructs of fiscal sensitivity and academic achievement in RSE than earlier research has [49,50].

Committed separation of the design of the 'long tail' nature of RSE is a critical academic and empirical need, especially in light of the fact that those public demands have the dual attributes of both heterogeneous privacy and universal externalities as other education categories. In this paper, we creatively draw parallels from the roles and functions of RBE on illiteracy to the public good nature of RSE on its unenrollment rate as the 'proxy variables' dividing this dual nature. Both of these results raise serious challenges to the study of Pulkkinen and Jahnukainen [51], which found that local stakeholders targeted the resources of special education differently with respect to the trade-off between the provision of special education and basic education, which needs to be resolved in the very near future.

Actually, an increasing number of researches supports the relationships among different expenditures and financing and RSE provision efficiency. Various researchers have argued that the interaction of different stakeholders in RSE could influence the results of RSE provision [8,21]. However, despite these theoretical and empirical consequences, the division and customization of specific subtypes and expenses of RSE based on its unique characteristics appear to be relatively scarce. Few perspectives that raise the idea argue that the combination of uniqueness and commonness of RSE and its corresponding dynamic correction interacted with other education products. Inspired from these perspectives, as well as from frameworks that have been widely applied to other areas such as business management, we propose a functional analysis of the dynamic correction and mediation effects of the 'long tail' and 'head' in public education as a whole. We also contribute to the fiscal implementation of divergent expenditures according to their various motivation effects on the correction.

There are limitations to our paper that need to be recognized and solved in future studies. First, the 'long tail' nature of RSE runs counter to the goal to fully financially support its inclusive satisfaction by governments, and other social sectors are encouraged to work on satisfying this goal. Given the 'long-tailed aggregator' roles of other social sectors (especially NPOs), which refers to the idea that the scattering and individualized distribution of RSE can also be satisfied through multiple customized provisions, there should be significant functions of

other social expenses on its correction. However, most empirical studies in our paper did not prove this channel by other social sectors. Moreover, it has been suggested that acts of relevant stakeholders follow a time-lag intertemporal channel, rather than lead to instant reactions, to recognize RSE demands and supply accordingly with the maximization of social benefits under political pressures. Thus, the RSE dynamic correction mechanism may be an alternative that can help RBE for the comprehensive popularization of universal education. Future research should compare these approaches to elicit the compensation effects independently rather than combine them.

Second, we have only examined the interaction and mediation effects of RSE with one 'head' demand RBE and 'long tail' RVE discretely. However, in addition to RBE and RVE, it has been proposed that other education categories serve a continuous multitude of functions, which include complementary or competitive roles with RSE. Hence, our results are restricted in the extent to which they center on the general theorized roles of RSE expenditures under a discrete context. Future study should be directed to check these successive education categories along the 'long tail' curve from the perspective of life course.

Third, while our study supports the catalytic role of different expenditures, especially fiscal incentives, it is plausible that the excessive investment and proportion of some specific expenditures do harm to the RSE correction under certain contexts. This paper provides a test of distortion and misallocation that the motivation of welfare and scholarship for RSE faculty members and students are obscure, and, with this awareness, future research needs to further investigate the time validity of the failure and devise more effective motivation mechanisms for those categories of expenses.

Fourth, in the current paper, the RSE dynamic correction mechanism was performed directly by the intertemporal convergence of uneducated RSE children; however, questions remain as to whether solely surveying a decreasing trend could serve as a conclusion for automatic correction. Moreover, although the current study addresses the time-lag effect of different expenditures by various social actors, we have yet to add the first-order time lag of *Unedu* into the model with respect to the issue of endogeneity. We also did not address the potential side effects of this underlying mechanism.

Fifth, given that there are large marginal effects of capital expenses, especially infrastructure construction associated with policy implementation, a limitation of our sensitivity check is that it did not consider the policy evaluation of other random shocks and rational expectations. Future research should isolate the function of the policy period that mixes with other interference terms. Given that RSE has its unique structure and nature, conducting RSE production and provision by small and private organizations does not guarantee that it will be economically profitable. Thus, our use of the 'long tail' theory in the field of public service strongly suggests that those organizations rigorously obey the customized rules and act as information filtering mechanisms. However, the problems of 'third-sector failure' may not ensure that our theory could generalize and apply across different organizations and evade information asymmetry and moral risk in the 'market' of public goods. Thus, the effect of correction may be due to the self-adjustment of social actors and the aggregation effects of expense, rather than to our division of different stakeholders. However, such concern for the negative effects of the 'long tail' should be minimized by the use of alternative measures of fiscal incentives, such as human capital intensity.

Given that existing studies imply that RSE demands are easy to ignore and mismatch, our results offer several indirect suggestions for how some expenditures (such as capital expenses) help to dynamically alleviate its imbalance and the extent of their marginal effects. In particular, we have proposed and found that the 'long tail' nature of RSE can directly influence the interaction and mediation of other education categories in the correction. These findings

suggest that the balance of RSE not only benefits from financial expenditures by governments but also may benefit from other social sectors, though not significantly in some models. Given that some categories of expense may be distorted and misallocated, we do not intend to advise relevant stakeholders to cancel these investments, which can be improved by better understanding and designing of the motivation channel for RSE faculty members and students. Rather, efforts should be directed towards fostering the formal instant recognition and provision mechanism, which can be achieved through cooperating with other social organizations acting as the 'long tail aggregator' and formalizing the network platform to share information, thus considering the dual nature of RSE together from implementing system detection. Moreover, given our findings that different subtypes of RSE have divergent performances associated with different expenses, the detailed and tailored pattern of 'cater to all disabilities' may be necessary for exclusive or comprehensive RSE schools. More importantly, our studies suggest that we could take a new perspective on the balance and equilibrium of RSE from system theory. In particular, rather than regarding the supply of RSE as originating solely statically, we may want to further investigate the dynamic cause of these interactions and determine whether they reflect the internal nature of RSE, such as scattering distribution and demand heterogeneity. Given that governments are likely to engage in the infrastructure construction of RSE as a way to restore balance, this paper has examined whether different expenses are reflective of different stakeholders' attempts to resolve the mismatch, as well as intertemporal tendencies to resort to broader investment in construction by addressing these issues.

## Note

1. 'Universal education' means that 85% of children could receive nine-year universal education (six-year primary school and three-year middle school).

2. The western and central provinces in China are Hebei, Shanxi, Inner Mongolia, Jilin, Heilongjiang, Anhui, Jiangxi, Henan, Hubei, Hunan, Guangxi, Hainan, Chongqing, Sichuan, Guizhou, Yunnan, Tibet, Shaanxi, Gansu, Qinghai, Ningxia, and Xinjiang. The eastern provinces are Beijing, Tianjin, Shandong, Jiangsu, Zhejiang, Shanghai, Fujian, Guangdong, and Liaoning.

## Author Contributions

**Resources:** Guangqin Li.

**Software:** Ji Luo.

**Supervision:** Guangqin Li.

**Writing – original draft:** Bowen Li.

**Writing – review & editing:** Guangqin Li, Ji Luo.

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
