## [Decision Letter · Decision Letter 0]

5 Oct 2020

PONE-D-20-21000

Latent but not Absent: The ‘Long Tail’ Nature of Rural Special Education and its Dynamic Correction Mechanism

PLOS ONE

Dear Dr. Li,

Thank you for submitting your manuscript to PLOS ONE. After careful consideration, we feel that it has merit but does not fully meet PLOS ONE’s publication criteria as it currently stands. Therefore, we invite you to submit a revised version of the manuscript that addresses the points raised during the review process.

In particular:

It would be interesting to analyze the expenditure items together, but not only as isolated items.The author should made explicit the set of control variables included in the alternative specifications.The author should consider, or as less discuss, the intertemporal implications of the expenditure programs.

We look forward to receiving your revised manuscript.

Kind regards,

María Carmen Díaz Roldán, Ph.D Economics

Academic Editor

PLOS ONE

Journal Requirements:

Reviewers' comments:

Reviewer's Responses to Questions

**Comments to the Author**

1. Is the manuscript technically sound, and do the data support the conclusions?

Reviewer #1: Yes

Reviewer #2: Yes

2. Has the statistical analysis been performed appropriately and rigorously? 

Reviewer #1: Yes

Reviewer #2: Yes

3. Have the authors made all data underlying the findings in their manuscript fully available?

Reviewer #1: Yes

Reviewer #2: No

4. Is the manuscript presented in an intelligible fashion and written in standard English?

Reviewer #1: Yes

Reviewer #2: Yes

5. Review Comments to the Author

Reviewer #1: • A very quick summary:

Authors includes rural special education (RSE) in the literature of ‘long tail’/’head’ public goods with the aim of identifying whether provision insufficiency may be impacted by different expenditures program composition carried out by governments and other social sectors as Non-Profits Organizations (NPOs). They use data from 30 provinces in China from 2003-2014 to estimate different models in which alternatively includes different categories and expenditure providers. Their findings indicate that there is an optimal trend/sequence to impact positively the provision insufficiency under analysis.

• Positive contributions:

1. ‘Long tail’ issues are difficult to analyze, in general. Not only because governments, academics and societies tend to focus on the ‘head’ issues but also because information may be more difficult to be collected. For the sake of concreteness, expenditure programs related to a wide variety of NPOs. In this regard, authors introduce a general conceptual framework to analyze different topics related to public economics and affecting a minor percentage of total population. So that, it may be revisited in future papers to analyze following this same approach.

2. Authors identify a ranking of expenditure categories in terms of their ability of reduce the provision insufficiency. From an alternative point of view, it may be useful for policymakers to identify the sequence and/or composition when designing expenditure policies related to RSE.

• Issues to be fixed/improved:

1. Regarding this paper, I have three main concerns:

(a) The different models treat expenditure categories as isolated items. On the contrary, when designing the expenditure program (public or private) is relevant both level and composition. Results obtained are useful in terms of the level assigned to each category, but they should be combined in the same model to confront the impact of all of them.

(b) Past expenditures obviously conditions the level of expenditure program assigned to a concrete rural special education school because, once you have built the school and you have provided the initial stock of inventories (capital expenditure) it could be more interesting to focus on other categories (welfare expenditures, among others). As far as I understood, all the items in the models are contemporaneous, indicating that the intertemporal relationships are not fully considered in the empirical analysis.

2. For the sake of transparency, it could be highly beneficial for a better understanding of the paper that authors make explicit the set of control variables included in the alternative specifications.

3. From a formal point of view, the text needs to be deeply revised to improve the drafting and all the small errors I have detected throughout the whole text.

Reviewer #2: In general, I consider that the paper is fine and meets the requirements to be published in its current state.

In particular, the topic is quite relevant, has a practical importance and has interest to the academic community. The paper is written in a clear and appropriate manner and it is well justified. The structure of the paper is correct and clear. In general, the literature review is correct as well as the theoretical framework. Both of them are useful to understand the research procedure and the methodology used. The study presents the results of original research and the methodology is performed to a high technical standard and is described in sufficient detail. Finally, conclusions are presented in an appropriate fashion and are supported by the data.

6. PLOS authors have the option to publish the peer review history of their article (what does this mean?). If published, this will include your full peer review and any attached files.

Reviewer #1: No

Reviewer #2: **Yes: **Beatriz Corchuelo

---

## [Author Response · Author response to Decision Letter 0]

14 Oct 2020

Response to Reviewers

First of all, we would like to thank reviewers for your insightful, constructive and helpful comments on our manuscript entitled “Latent but not Absent: The ‘Long Tail’ Nature of Rural Special Education and its Dynamic Correction Mechanism”. We have carefully considered and addressed all the comments and made necessary revisions in the revised manuscript. We provide a point-by-point response to the reviewers’ comments below.

The points raised by the reviewers are written in bold font, whereas our responses are shown in normal font, and the quotation of the revised manuscript is shown in italic font. 

Reviewer #1:

Authors includes rural special education (RSE) in the literature of ‘long tail’/’head’ public goods with the aim of identifying whether provision insufficiency may be impacted by different expenditures program composition carried out by governments and other social sectors as Non-Profits Organizations (NPOs). They use data from 30 provinces in China from 2003-2014 to estimate different models in which alternatively includes different categories and expenditure providers. Their findings indicate that there is an optimal trend/sequence to impact positively the provision insufficiency under analysis.

Positive contributions:

1. ‘Long tail’ issues are difficult to analyze, in general. Not only because governments, academics and societies tend to focus on the ‘head’ issues but also because information may be more difficult to be collected. For the sake of concreteness, expenditure programs related to a wide variety of NPOs. In this regard, authors introduce a general conceptual framework to analyze different topics related to public economic and affecting a minor percentage of total population. So that, it may be revisited in future papers to analysis following this same approach.

2. Authors identify a ranking of expenditure categories in terms of their ability of reducing the provision insufficiency. From an alternative point of view, it may be useful for policymakers to identify the sequence and/or composition when designing expenditure policies related to RSE.

Response:

Thank you very much for taking time out of your busy schedule to review our manuscript. At the same time, thank you very much for your positive comments on this article. Indeed, as you said, it is very difficult to investigate the "long tail" effect. In the manuscript, we first analyze the long tail effect of rural special education (RSE) through a general theoretical model, and then use the panel data of 30 provinces in China from 2003 to 2014 to empirically investigate the long tail effect of different types of expenditure in RSE and its dynamic correction mechanism. However, there are still many problems in our manuscript. We have done our best to revise the manuscript, but if any additional revision is needed, we will certainly do so under your directions.

Issues to be fixed/improved:

1. Regarding this paper, I have three main concerns:

(a) The different models treat expenditure categories as isolated items. On the contrary, when designing the expenditure program (public or private) is relevant both level and composition. Results obtained are useful in terms of the level assigned to each category, but they should be combined in the same model to confront the impact of all of them.

Response:

Thank you very much for your kind, thoughtful and valuable comments. According to your suggestion, we have revised the manuscript. Please see pages 25-26 in the revised manuscript: 

…

As the benchmark model, we first put all categories of expenditures combined-together in the same model to confront the impact of all of them (fixed effect). We set four variables to stand for the total administrative expense, the total welfare expense, the total scholarship expense and the total capital expense (public and private, or financial and social added together). The results are as show below:

Table 1. The Results of Total Expense on RSE Correction

Model Fe

DV Unedu

Total-Administrative -2.130*

 (-1.967)

Total-Welfare 2.889

 (1.567)

Total-Scholarship -7.641*

 (-2.002)

Total-Capital -3.253**

 (-2.333)

Year No

Fixed effect Yes

N 326

We found that for the base model of total expense, the coefficients of Total-Administrative, Total-Scholarship and Total-Capital are significantly negative (at least 10% level), which means they have significant alleviation effect on RSE correction. But the coefficient of Total-Welfare is not significant.

…

(b) Past expenditures obviously conditions the level of expenditure program assigned to a concrete rural special education school because, once you have built the school and you have provided the initial stock of inventories (capital expenditure) it could be more interesting to focus on other categories (welfare expenditures, among others). As far as I understood, all the items in the models are contemporaneous, indicating that the intertemporal relationships are not fully considered in the empirical analysis. 

Response:

Thank you very much for your kind, thoughtful and valuable comments. Indeed, we didn’t discuss the intertemporal relationship of expenditures in the manuscript. According to your suggestion, we have added a part of the revised manuscript to discuss the intertemporal impact. Please see pages 39-40 in the revised manuscript:

…

5.3 Intertemporal Results of Expenditures

As the demand-supply unbalanced degree of RSE may have self-correction dynamic mechanism intertemporal, we add the first-order lag value of Unedu as another independent variable in the model. Meanwhile, considering the endogeneity issue, we set the second-order lag value of Unedu as the instrumental variable (IV) of first-order lag value. Besides, compared with social expense on RSE, the effect of financial expense may have path dependence (because of the larger implementation cost and time of bureaucracy). Hence, we also add the first-order lag value of the service budget, GDP per capita, financial budget and population density as independent variables to set panel IV model (1). We first set the first-order difference for panel data, and then use IVs. 

The choice and use of IV could only alleviate but not totally overcome endogeneity problem. In order to further analysis, the intertemporal dynamic balance of RSE, we set dynamic panel DIF-GMM model (2) and SYS-GMM model (3). We choose the second-order lag value of Unedu as dependent variables, and use third-order lag value of Unedu at most as IVs. Considering the enrollment number, financial expense and social expense of RSE have relations to the real demand and other institutional factors on the last period, these are better to be set as antecedent variables due to their unobservability and reflect on the first-order lag random error. We set the current and first-order lag value as antecedent variables and second-order and third-order lag value as IVs. What’s more, the school construction area has endogeneity due to its relations to financial expense on infrastructure depreciation degree. Hence, we set it as endogenous variable and use second-order and third-order lag value as IVs. All other control variables are exogenous variables and two-step method and robust standard deviation are used. The results are as below:

Table The Intertemporal Results of Expenditures

 (1) (2) (3)

DV Unedu (Current)

 Panel IV DIF-GMM SYS-GMM

Unedu (-1, Did-Lag) -0.211** 

 (-2.105) 

Unedu (-1, Lag) -0.452*** -0.180**

 (-3.248) (-2.533)

Unedu (-2, Lag) -0.210 -0.016

 (-1.571) (-0.252)

Different Expense (current) Yes Yes Yes

Different Expense (-1, Did-Lag) Yes Yes Yes

Different Expense (-1, Did) Yes Yes Yes

Different Expense (-1, Lag) Yes Yes Yes

Control (-1, Did-lag) Yes Yes Yes

Control (-1, Did) Yes Yes Yes

Control (-1, Lag) Yes Yes Yes

Control (current) Yes Yes Yes

N 272 271 304

Note: -1 and -2 means the first-order and second-order lag value. Did means difference. Lag means lag value. Current means current value.

The exogeneity test of Panel IV (1) rejects all IVs are exogeneous at 1% level, hence DIF-GMM and SYS-GMM is necessarily needed to correct the endogeneity. The autocorrelation test of model (2) and (3) prove the first-order autocorrelation for the disturbance term, but no second-order autocorrelation exists. The over-identification test shows the p-value is 1, which couldn’t reject all IVs are effective. The Hansen-statistics also couldn’t reject the exogenicity of all IVs.

The results show the coefficients of the first-order lag value of Unedu is significantly negative, which proved the intertemporal dynamic self-correction of RSE. But the coefficients of second-order lag value are not significant. Hence, this self-correction only happened between successive periods in time series. For the dynamic imbalance trend between two periods or more, the influence is weak. 

As to school construction area as the endogenous variable (didn’t show in the table), the coefficients of current and first-order lag value are significantly negative, which proved its intertemporal effect due to the relatively long periodicity of infrastructure construction. It is also similar with other expense, as the coefficients of first-order lag is smaller than the current value, which proved this decreasing trend in time series.

…

What’s more, relative expenditure programs and index setting (like minimum dropout rate of RSE) have a positive effect on the dynamic adjustment and correction of ‘long tail’ rural public service. On the basis of full investigation and evaluation to understand the real needs of rural areas, sound and dynamic institutions should be implemented to activate supply sectors to lift governance level, meet unsatisfied demands and strengthen supervision and evaluation. 

…

2. For the sake of transparency, it could be highly beneficial for a better understanding of the paper that authors make explicit the set of control variables included in the alternative specifications.

Response:

Thank you very much for your kind, thoughtful and valuable comments. Thank you very much for your reminder. Please see page 23 in the revised manuscript:

…

which include the rural per capita income, the rural per capita amount of social cultural organizations, the rural computer room area for special education, the budget expenditure of special education per RSE student, marketization degree, the budget expenditure of basic education per rural normal student and rural population (All models are the same).

…

3. From a formal point of view, the text needs to be deeply revised to improve the drafting and all the small errors I have detected throughout the whole text.

Response:

Thank you very much for pointing out the errors in the whole text. We have done our best to revise the manuscript, but if any additional revision is needed, we will certainly do so under your directions.

 

Reviewer #2: 

In general, I consider that the paper is fine and meets the requirements to be published in its current state. In particular, the topic is quite relevant, has a practical importance and has interest to the academic community. The paper is written in a clear and appropriate manner and it is well justified. The structure of the paper is correct and clear. In general, the literature review is correct as well as the theoretical framework. Both of them are useful to understand the research procedure and the methodology used. The study presents the results of original research and the methodology is performed to a high technical standard and is described in sufficient detail. Finally, conclusions are presented in an appropriate fashion and are supported by the data.

Response:

Thank you very much for taking time out of your busy schedule to review our manuscript. We have done our best to revise the manuscript based on the suggestions of other reviewers, but if any additional revision is needed, we will certainly do so under your directions.

 

Editor: 

1. It would be interesting to analyze the expenditure items together, but not only as isolated items.

2. The author should have made explicit the set of control variables included in the alternative specifications.

3. The author should consider, or as less discuss, the intertemporal implications of the expenditure programs.

Response:

Thanks to the editor for summarizing the core issues of the revision, we made a comprehensive revision to the corresponding parts according to the reviewer's comments, and comprehensively revised the text expression problems existing in the full text. But if any additional revision is needed, we will certainly do so under your directions.

---

## [Editor Report · Decision Letter 1]

26 Oct 2020

Latent but not Absent: The ‘Long Tail’ Nature of Rural Special Education and its Dynamic Correction Mechanism

PONE-D-20-21000R1

Dear Dr. Li,

We’re pleased to inform you that your manuscript has been judged scientifically suitable for publication and will be formally accepted for publication once it meets all outstanding technical requirements.

Kind regards,

María Carmen Díaz Roldán, Ph.D Economics

Academic Editor

PLOS ONE

Additional Editor Comments (optional):

The author has satisfactorily addressed the corrections required by the reviewers.
---

## [Editor Report · Acceptance letter]

11 Nov 2020

PONE-D-20-21000R1 

Latent but not Absent: The ‘Long Tail’ Nature of Rural Special Education and its Dynamic Correction Mechanism 

Dear Dr. Li:

I'm pleased to inform you that your manuscript has been deemed suitable for publication in PLOS ONE. Congratulations! Your manuscript is now with our production department. 

Kind regards, 

on behalf of

Dr. María Carmen Díaz Roldán 

Academic Editor

PLOS ONE